# Orthodontic Internal Resorption Assessment in Periodontal Breakdown—A Finite Elements Analysis (Part II)

**DOI:** 10.3390/healthcare11192622

**Published:** 2023-09-25

**Authors:** Radu Andrei Moga, Ada Gabriela Delean, Stefan Marius Buru, Mircea Daniel Botez, Cristian Doru Olteanu

**Affiliations:** 1Department of Cariology, Endodontics and Oral Pathology, School of Dental Medicine, University of Medicine and Pharmacy Iuliu Hatieganu, Str. Motilor 33, 400001 Cluj-Napoca, Romania; 2Department of Structural Mechanics, School of Civil Engineering, Technical University of Cluj-Napoca, Str. Memorandumului 28, 400114 Cluj-Napoca, Romania; marius.buru@mecon.utcluj.ro (S.M.B.); mircea.botez@mecon.utcluj.ro (M.D.B.); 3Department of Orthodontics, School of Dental Medicine, University of Medicine and Pharmacy Iuliu Hatieganu, Str. Avram Iancu 31, 400083 Cluj-Napoca, Romania; olteanu.cristian@umfcluj.ro

**Keywords:** internal orthodontic resorption risks, periodontal breakdown, orthodontic movements, finite elements analysis, failure criteria

## Abstract

This finite elements analysis (FEA) assessed the accuracy of maximum shear stress criteria (Tresca) in the study of orthodontic internal surface resorption and the absorption–dissipation ability of dental tissues. The present study was conducted over eighty-one models totaling 324 simulations with various bone loss levels (0–8 mm), where 0.6 N and 1.2 N were applied in the intrusion, extrusion, rotation, tipping, and translation movements. Tresca criteria displayed localized high-stress areas prone to resorption for all situations, better visible in the dentine component. The internal resorptive risks are less than external ones, seeming to increase with the progression of the periodontal breakdown, especially after 4 mm. The internal and external surface high-stress areas are strictly correlated. The qualitative stress display for both forces was almost similar. The rotation and tipping displayed the highest resorptive risks for the pulp chamber, decreasing with bone loss. The resorptive risks seem to increase along with the progression of periodontal breakdown if the same applied force is kept. The dentine resemblance to ductile based on its high absorption–dissipation ability seems correct. Tresca seems to supply a better predictability of the prone-to-resorption areas than the other failure criteria.

## 1. Introduction

The resorptive potential of orthodontic treatment is acknowledged as a common, inevitable, and unpredictable side effect in both the periodontium and tooth [1,2]. The orthodontic movement is initiated by circulatory disturbances at the periodontal ligament (PDL) level due to variations in the physiological maximum hydrostatic pressure (MHP) [2,3].

The effects of the orthodontic forces in PDL are various levels of ischemia (the higher the force, the stronger the level of ischemia seemed to be), stimulating the initiation of movements, ischemic circulatory disturbances in dental pulp, neuro-vascular bundles (NVB), and further periodontal loss (strictly correlated with the levels of the already present bone loss) [3,4,5].

In the tooth, the effects of the orthodontic force are related mainly to the appearance of resorptive areas/lacunae both on the external and internal surface of the root, root canals, and pulp chamber (the higher the applied force, the faster and more extensive the resorptive areas developed) [6,7,8,9].

There are close correlations and relationships between the amount of the applied orthodontic force and the maximum hydrostatic pressure (MHP) in the PDL and dental pulp–NVB tissues. The MHP was reported to be 12.8–16 KPa (about 80% of the systolic pressure), and it is advised that this not be exceeded (to avoid ischemic loss), while it is recommended that the minimum hydrostatic pressure of 4.7 KPa be surpassed in order to trigger the movement [3,4,5,6,7,10,11,12]. There are numerous reports regarding the optimal applied force of about 1 N (light forces of 0.5–1 N/approx. 50–100 gf), triggering the orthodontic movements but without significant ischemic and resorptive risks [2,4,5,6,8,10,11,13,14]. Opposingly, other studies adopt as optimal strengths amounts of force much higher without reporting any major resorptive/ischemic/necrotic risks 0.28–3.31 N/approx. 28–331 gf [15,16,17], considering that the light forces do not effectively trigger the alveolar bone remodeling processes [6].

Nevertheless, it is acknowledged that the high amounts of applied force produce localized points/areas of high pressure (where the stresses produced by the applied force are high) over the external and internal surface of the tooth, with the results of the internal and external orthodontic resorptive processes [6,7,18] continuing another 4 weeks after the force action has stopped [1].

The prevalence of the resorptive process (affecting both the root and crown [2,6,14]) due to orthodontic causes was reported to be variable, with reports between 0.02–2.3% (external cervical) and 1–5% (external radicular) [2], and 1–51.5% (internal radicular) [2], up to 20–100% [1,7,14]. Usually, the resorptions are unpredictive isolated localized areas/lacunae of various depths and surface extents, with no clinical symptomatology appearing after two–four weeks after applying the orthodontic force [19,20]. However, symptoms and signs of acute/chronic pulpitis could be present if the ischemic disturbances are important [2]. The mechanism is not completely understood and studied; it seems that the source is the ischemic disturbances affecting the MHP, triggering resorptive processes until the pressure source is removed [2,14]. There are little data available regarding the internal root resorption and no finite elements studies (FEA) were found regarding this issue [2].

The internal root resorption seems to be initiated along the root canal with the progressive destruction of the dentine (damage to the odontoblasts and unmineralized pre-dentine) due to orthodontic treatment pressures and localized inflammation of the dental pulp tissue of ischemic orthodontic origin [2]. The process consists of two phases, the transient phase with a self-limiting resorption, and the progressive phase with ischemic necrosis triggering bacterial resorptive activity [2]. There are also two types of resorptions: inflammatory internal resorption (i.e., loss of intra-radicular dentine without any deposition in the resorptive lacunae), and replacement internal root resorption (i.e., loss of intra-radicular dentine replaced by metaplastic bone/cementum-like hard tissues) [2].

The clinical treatment of the internal root resorption depends on the results of the extent of destruction and whether there are present signs of pulp sufferance demanding endodontic root canal treatment and/or regenerative endodontic procedures. If the perforations are present, the endodontic regenerative procedures include calcium hydroxide and/or antibiotic paste for a variable period of 0.5–3 months followed by MTA filling [2,9].

The finite elements analysis (FEA) is a well-known study method allowing individual analysis of each component of a structure, especially useful in extremely small structures that cannot be otherwise studied [4,5,8,9,10,11,12,21].

The in vivo dental tissues (tooth and surrounding periodontium in particular) are particularly suited for the use of FEA due to their small anatomical dimensions and complex anatomy. FEA is an exact mathematical method, widely used in the engineering field, providing accurate results if the employment conditions and the input data are correct [4,5,8,9,10,11,12,21,22].

In dentistry, FEA studies have often reported debatable results, sometimes contradicting the available clinical data, making this useful study method the object of mistrust due to the misunderstanding of the yielding materials theory (which relies on the failure criteria design). Nonetheless, there are reports that if adequate material-based failure criteria and anatomical correct input data are used, then the results will be accurate and correlate with the in vivo clinical data [4,5,8,10,11,12,22].

When the dental tissues are FEA-studied, their micro-architecture and physical properties must be acknowledged, as well as their biomechanical behaviors [3,13,22,23,24,25]. When subjected to an amount of force, the tooth components and surrounding periodontium display a remarkable ability of absorption–dissipation, such that only a reduced amount of stress directly affects the surrounding periodontium and dental pulp–NVB [3,4,5,8,10,11,12,23,24,25,26,27]. This biomechanical ability of absorption–dissipation corresponds to ductile materials; otherwise, the dental tissues being reported resemble ductile materials (but with a certain brittle flow mode) [3,4,5,8,10,11,12,23,24,25,26,27].

The debatable reports of the FEA dental studies are due more to the misemployment of the failure criteria and less because of the input data [3,8,10,11,21]. The failure criteria are material-based designed (each material type poses a certain biomechanical behavior under stress—the yielding theory) [3,8,10,11,21]. The main difference is due to their different deformations under stress, the ductile (e.g., steel, rubber) suffers from elastic/plastic deformation with variable degrees of reversibility before fracture/destruction, while brittle materials (e.g., stone, glass) suffer directly from fracture/destruction [3,8,10,11,21]. All dental tissues (dentine, cementum, dental pulp, NVB, and PDL) are considered to resemble ductile, except for the enamel, which is considered to resemble brittle [4,5,8,10,11,12,23,24,25,26,27]. Nevertheless, the enamel is only a small % of the tooth structure; thus, the tooth biomechanical behavior resembles that of ductile [4,5,8,10,11,12,23,24,25,26,27]. Despite multiple reports and clinical evidence [4,5,8,10,11,12,23,24,25,26,27] regarding the ductility resemblance of dentine, there are still studies [22] that consider dentine to resemble a brittle solid, without further discussion of the issues, suggesting that both maximum principal and von Mises criteria should be used [22]. Moreover, other studies actively employed hydrostatic stresses and reported them to be the only adequate criteria, reporting different results from one study to another and for the same issues [15,16,17,28,29].

There are many FEA studies of PDL, while for the external root resorption, only a few were available, and none for the external and internal root resorption in periodontal breakdown. No studies related to the resorption process (neither external nor internal) were found for the periodontal breakdown despite various bone levels being found in orthodontic patients. Moreover, no information about the amount of safely applied orthodontic force to avoid orthodontic resorption or absorption–dissipation ability was found. These studies used various failure criteria (von Mises [9,21,30,31], maximum principal S1 tensile stress/minimum principal S3 compressive stress [21,30,31,32,33], and hydrostatic pressure [6,7,15,16,17,28,29,30,31,34]), without any correlation with the type of analyzed material, MHP, ischemic risks, and reporting results that sometimes contradicted the clinical data or the previously reported results [3,4,5,8,10,11,12]. However, a series of recent FEA studies [3,4,5,8,10,11,12], comparing the different failure criteria, arguing the material-based type failure criteria selection, and correlating the MHP and ischemic risks with the biomechanical behavior of tooth, PDL, dental pulp, and NVB, reported Tresca failure criteria as providing more accurate results than that of the other failure criteria.

Tresca failure criteria state that failure occurs in a material when the absolute value of maximum shear stress reaches the shear stress at yielding and is consistent with the failure being independent of hydrostatic stress.

Clinical external resorption studies have reported that compression and pressure surfaces (i.e., stress distribution) were more prone to resorptive risks [20], while the severity of the resorptive process (depth and extent) was correlated with the amount of applied force [7,20,34]. Based on these observations, the need to respect the anatomical accuracy of the analyzed 3D models (the lacunae location is dependent on the anatomical morphology and curvatures of the tooth) becomes evident [34]. Another report [34] suggested the importance of the correlation between stress distribution, root morphology, and the resorptive lacunae, with lesser importance given to the amount of the applied force. This approach reinforces the concept that the shear stress induces in the root and coronal surfaces local pressures, producing localized resorptive lacunae, while in the dental pulp and PDL, various levels of ischemia are produced, triggering both the orthodontic movement, the resorptive process, and a further tissue loss. Practically, the resorptive process depends on the amount of applied force strictly correlated with the absorption–dissipation ability of the tooth and the exceeding of the 4.7 KPa of minimum hydrostatic pressure but without further exceeding the 16 KPa of the MHP, as well as the periodontal support level [4,5,8,12,34]. Furthermore, the high absorption–dissipation ability of the tooth for internal shear stress was clinically reported through the prevalence of internal resorption vs. the external one [1,2,6,7,14], and through FEA reports [4,5,8,12] about higher amounts of shear stresses reaching the PDL than those reaching the dental pulp and NVB.

FEA studies are the only methods to individually assess the stress distribution and display in living tissues with accurate results (as in the engineering field) if the requirements for the use of the criteria are met [3,4,5,8,10,11,12,21,22]. The few studies [7,20,28,29,30,31,34] that assessed the external root resorption by combining an in vivo–in vitro experiment and an FEA analysis of an intact periodontium, reported FEA results that did not accurately match the clinical report. Moreover, a non-internal resorption study was found probably due to the difficulty in the analysis of this type of lesion, despite its high reported prevalence of up to 51.1% [2]. Thus, the FEA analysis employing a ductile materials failure criterion (maximum shear stress—Tresca [4,5,8], more adequate for the study of dental tissues) was considered an adequate approach in the study of the internal resorption issue.

This FEA analysis aimed to investigate the areas more exposed to the risks of orthodontic internal resorption during 0–8 mm of periodontal breakdown under 0.6 N/60 gf and 1.2 N/120 gf and under various movements, as well as the absorption–dissipation ability of the tooth. Additionally, if there are significant differences between the biomechanical behavior of the two forces, they were also assessed.

## 2. Materials and Methods

The current FEA analysis represents a stepwise study of a more comprehensive research (clinical protocol nr.158/02.04.2018) of the biomechanical behavior of the tooth and surrounding support tissues under orthodontic movements during the horizontal periodontal breakdown [3,4,5,8,10,11].

This analysis involved a number of 324 simulations on eighty-one anatomically correct 3D models from nine patients (mean age of 29.81 ± 1.45 years, 4 males and 5 females, oral informed consent).

This research examined a large number of patients, but a rather restrictive inclusion criteria reduced their number to nine. These inclusion criteria were chosen for reducing the biomechanical uncertainties (i.e., intact mandibular arch, no malposition, intact teeth, no root canal endodontic treatment, no filling, non-inflamed periodontium, moderate/reduced bone loss in the cervical third, orthodontic treatment, regular periodic checks). The exclusion criteria were incomplete arches, malposition teeth, restored teeth, large amount of bone loss, and inflamed periodontium. Thus, the sample size was nine (nine patients; nine models/patient; a total of 81 3D models and 324 FEA simulations), as opposed to current FEA studies analyzing a sample size of one (one model of one patient and few FEA simulations) because of the 3D model creation. It must be emphasized that most of the previous FEA studies [7,9,15,16,17,20,22,23,26,27,28,29,30,31,32,33,34] used for comparison with the herein results used a sample size of one (one patient and one model), with the exception of our earlier one [3,4,5,8,10,11] (a sample size of nine).

The starting point for the 3D models was the CBCT (cone beam computed tomography, ProMax 3DS, Planmeca, Helsinki, Finland, voxel size 0.075 mm) examination of the mandibular arch (i.e., premolars and molars). The manual image reconstruction was performed by a single practitioner employing the AMIRA 5.4.0 software (Visage Imaging Inc., Andover, MA, USA). On each radiological image/slice the anatomical components were identified (based on the Hounsfield grey shade units) and selected. Thus, the enamel, dentine, bracket, periodontal ligament, dental pulp neurovascular bundle, and cortical and trabecular bone were found and selected. Due to difficulties in identifying the cementum component and the similar physical properties with the dentine (Table 1), the cementum was reconstructed as dentine (the radicular dentine/cementum component). All these components were merged in a 3D model (for each of the nine patients, e.g., Figure 1 and Figure 2) with limited varied cervical third bone loss.

The periodontal ligament has an average thickness of 0.15–0.22 mm and was included in the apical third of the neurovascular bundle of the dental pulp. The first premolar and the two molars were replaced by cortical and trabecular bone. The missing bone and PDL were as much as possible closely anatomically reconstructed, obtaining nine models with an intact periodontium with the second lower premolar.

All models were then subjected to smoothing and refinement processes, obtaining 3D models with a total of 5.06–6.05 million C3D4 tetrahedral elements, 0.97–1.07 million nodes, and global element size of 0.08–0.116 mm (e.g., Figure 1 and Figure 2). All models had surface irregularities since the reconstruction was manually conducted. However, the internal algorithm of the reconstruction and FEA software does not allow a further step when there are errors, anomalies, and/or irregularities that could interfere with the biomechanical behavior. Thus, despite the presence of surface irregularities, those are in non-essential areas, while the stress display areas are quasi-continuous. Moreover, the FEA software allows a mesh testing for evaluating the total number of errors and warnings, resulting in no errors and only a limited number of element warnings (e.g., Figure 1 and Figure 2: 264 element warnings (representing 0.0043%) for the entire model of 6.05 million C3D4 elements; 63 element warnings (0.00677%) for the 930,023 elements of the tooth, bracket, and PDL; 26 element warnings (0.00459185%) for the 566,221 elements of the radicular dentine/cementum and coronal dentine; and 17 element warnings (0.0141469%) for the 120,168 elements of the enamel and bracket).

Each of the intact periodontium nine models was subjected to gradual horizontal PDL and bone reduction of 1 mm, simulating a horizontal periodontal breakdown process of 0–8 mm of loss, obtaining a total of 81 models.

The assumed boundary conditions were isotropy, homogeneity, and linear elasticity as in the other studies found in the scientific flow. Moreover, the assumptions were considered acceptable since under small loads (around 1 N/100 gf), the biomechanical movements are extremely small, and all tissues display linear elasticity.

The FEA analysis totaling 324 simulations was conducted using the ABAQUS6.13-1 software (Dassault Systèmes Simulia Corp., Maastricht, The Netherlands). The failure criteria were Tresca maximum shear stress specially designed for describing the biomechanical behavior of ductile resemblance non-homogeneous materials, considered to be more adequate for the study of dental tissues than other criteria [8,10,11].

The applied forces at the bracket level (e.g., Figure 1) were 0.6 N/approx. 60 gf and 1.2 N/approx. 120 gf, simulating five orthodontic movements (extrusion, intrusion, tipping, rotation, and translation). Those forces were chosen not only because they are often used in clinical practice but also to establish correlations with previous analyses [4,5,8] of PDL and dental pulp and NVB, thus improving the knowledge regarding the biomechanical behavior of teeth subjected to periodontal breakdown.

**Figure 1 healthcare-11-02622-f001:**
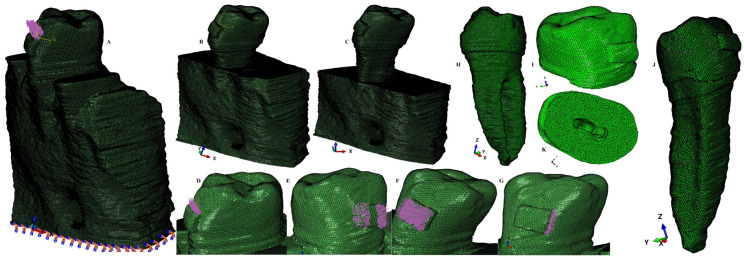
Mesh model: (**A**) 2nd lower right premolar model with intact periodontium and applied vector for extrusion, (**B**) mesh model with 4 mm bone loss, (**C**) mesh model with 8 mm bone loss, (**D**) applied vector for intrusion, (**E**) applied vector for rotation, (**F**) applied vector for tipping, (**G**) applied vector for translation, (**H**) mesh model of radicular dentine/cementum and coronal dentine, (**I**) enamel and bracket components, (**J**) 2nd mandibular premolar with applied bracket, (**K**) section of tooth with pulp-camber and root canals.

**Figure 2 healthcare-11-02622-f002:**
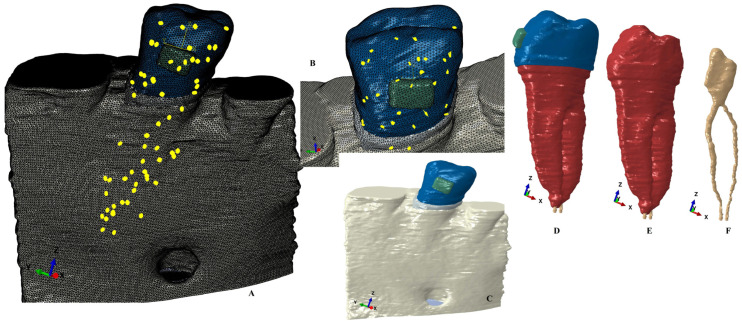
Mesh model: (**A**) mesh model grid with intact periodontium and 63 elements warning for the tooth, bracket, and PDL, (**B**) 39 elements warnings for tooth and bracket, (**C**) model with no bone loss without the mesh grid, (**D**) tooth with bracket dental pulp and NVB without the mesh grid, (**E**) radicular dentine/cementum and coronal dentine components with NVB, (**F**) dental pulp and NVB.

**Table 1 healthcare-11-02622-t001:** Elastic properties of materials.

Material	Young’s Modulus, E (GPa)	Poisson Ratio, ʋ	Refs.
Enamel	80	0.33	[3,4,5,8,10,11]
Dentine/Cementum	18.6	0.31	[3,4,5,8,10,11]
Pulp	0.0021	0.45	[3,4,5,8,10,11]
PDL	0.0667	0.49	[3,4,5,8,10,11]
Cortical bone	14.5	0.323	[3,4,5,8,10,11]
Trabecular bone	1.37	0.3	[3,4,5,8,10,11]
Bracket (Stainless Steel)	190	0.265	[3,4,5,8,10,11]

The FEA supplied qualitative color-coded (projections of the shear stress areas) and quantitative results (average shear stress). The color-coded results were red-orange (maximum shear stress prone to high resorptive risks), yellow (high shear stress having a moderate to high risk of resorption), yellow-green (moderate resorption risks), and blue-green (reduced resorptive risks). These color-coded areas were displayed on both the entire tooth structure (including pulp and bracket) and on the radicular dentine/cementum and coronal dentine components (on both internal and external surfaces), analyzing the apical, middle, and cervical thirds and their sides/walls (vestibular, lingual, mesial, distal). The quantitative average shear stresses were correlated with the quantitative shears stress for PDL [5] and dental pulp–NVB [4] (under 0.6 N and 1.2 N of applied force) for assessing the ability of tooth structure to absorb–dissipate [8] the stresses and highlight the areas more exposed to risk of resorption.

## 3. Results

The present FEA analysis employed the maximum shear stress failure criteria over eighty-one models totaling 324 simulations (e.g., Figure 3, Figure 4, Figure 5, Figure 6, Figure 7 and Figure 8 and Table 2 and Table 3). As expected, no gender and age-related differences were seen.

The simulations assessed both qualitatively and quantitatively the shear stress in the entire tooth structure and radicular/root dentine/cementum and coronal dentine components, describing and displaying the same biomechanical behavior with complementary qualitative (tooth structure vs. dentine/cementum and coronal components) and quantitative (lower dentine/cementum vs. tooth structure) results. Because of the absence of interference with other anatomical components (present in the tooth structure), by independently analyzing the radicular dentine/cementum and coronal dentine components, a more correct, concise, and precise representation of the areas of maximum, high, and moderate shear stresses to undergo a potential internal resorption was possible (e.g., Figure 3, Figure 4, Figure 5, Figure 6, Figure 7 and Figure 8).

From the qualitative point of view, the maximum shear stress prone to higher resorptive risks was color-coded in red-orange, while the yellow and yellow-green areas signaled localized moderate resorptive risks. In the radicular dentine/cementum and coronal dentine components, the red-orange localized areas were visible on the external side of the root and crown (for all movements, bone levels, and loads), while the internal surface always displayed only mild resorptive risks (yellow and yellow-green areas) (e.g., Figure 3, Figure 4, Figure 5, Figure 6, Figure 7 and Figure 8). The tooth structure displayed red-orange areas of higher external resorptive risks in the rotation, translation, and tipping movements (bodily movements) after 4 mm of bone loss, significantly more observable in the radicular dentine/cementum and coronal dentine. Tresca criteria supplied biomechanically complementary correct and correlated stress display areas on both the tooth structure, and external and internal surfaces of radicular dentine/cementum and coronal dentine components, offering an adequate display/localized areas of the maximum, high, and moderate stress areas facing higher and moderate/mild resorptive risks. Both applied forces (0.6–1.2 N) displayed an almost similar shear stress display and extent in all analyzed structures for all movements and bone loss levels.

From the quantitative point of view, the average amount of stress displayed in the internal surface of the radicular dentine/cementum component was lower when compared with the external surface and tooth structure, for all movements and forces (Table 2). Nevertheless, the internal and external shear stress displayed in the coronal dentine component (directly influenced by the action of orthodontic forces) was almost similar, but 4–5 times lower when compared with the stress around and under the bracket area of the tooth structure.

In all simulations, the amounts of shear stress displayed a progressive increase correlated with the progression of periodontal breakdown (Table 2 and Table 3). The average quantitative stress doubled when 1.2 N was applied, compared with 0.6 N (Table 2 and Table 3). With the progression of bone loss, the stress in the internal surface of the radicular dentine/cementum component reached a maximum of 4–5 times higher in the apical and middle third for 8 mm of loss when compared to intact periodontium. A much higher biomechanical difference between 0 and 8 mm of loss (of 7–11 times stress increase) was observed for the external surface of the radicular dentine/cementum component and tooth structure. Nevertheless, all the quantitative stress results (Table 2 and Table 3) were lower than the 29–73.1 MPa reported as maximum shear stress for dentine [23].

### 3.1. Extrusion

In the extrusion movement, from a biomechanical point of view, the tooth structure displayed visible areas of higher stress after 4 mm of bone loss on the vestibular and lingual sides (e.g., Figure 3A,B). In the radicular dentine/cementum component, the extrusion movements displayed the localized areas of the maximum shear stress (color-coded red-orange, and prone to higher resorptive risks) for both intact and reduced periodontium in the external surface (mainly vestibular, especially from 0 to 4 mm of bone loss and correlated with the level of periodontal breakdown) (e.g., Figure 3C,D). Mild/moderate resorptive risks (yellow and yellow-green areas) were also visibly displayed on both the external and internal surfaces of the radicular dentine/cementum. The moderate resorptive risks were externally visible on the vestibular, and lingual entire sides (in cervical, middle, and apical thirds), while for the internal surface, the pulp chamber seemed less affected (the vestibular side/wall) than the vestibular root canal (cervical, middle, and apical thirds) (e.g., Figure 3E–J).

Quantitatively, the shear stress in the internal surface of the radicular dentine/cementum was less (half in the apical, middle, and cervical thirds) than the external surface values and of the entire tooth structure (Table 2). However, the internal shear stress in the apical and middle third of the radicular dentine/cementum doubled at 8 mm of loss compared with the intact periodontium, while in the coronal dentine, it remained almost the same. The highest shear stress for the coronal dentine component was displayed in the intact periodontium around and under the bracket with a visible interest of the vestibular side/wall of the pulp chamber (e.g., Figure 3C,D,J). Nevertheless, with the progression of the bone loss, the shear stress decreased, showing less interest in the pulp chamber. The mild internal resorptive risk areas (yellow-green) displayed mainly in the vestibular root canal for 0–8 mm (and lingual root canal from 4 to 8 mm) of bone loss seemed to be more extensive as the periodontal breakdown progressed (no bone loss—cervical third; 4 mm loss—middle and apical third; 8 mm loss—entire apical third).

**Figure 3 healthcare-11-02622-f003:**
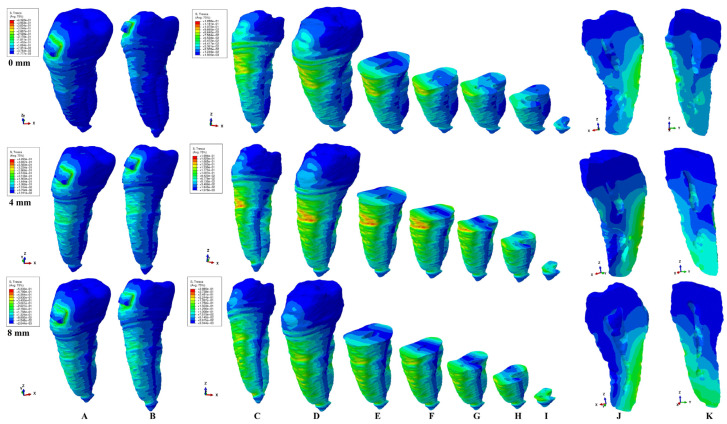
A 0.6 N of extrusion—comparative shear stress display between the tooth structure (**A**,**B**) and the radicular dentine/cementum and coronal dentine components ((**C**,**D**) dentine component, (**E**–**I**)—various horizontal radicular sections in cervical, middle, and apical thirds, (**J**) vertical section of the dentine component viewed from distal side, (**K**) vertical section of the dentine component viewed from lingual side).

### 3.2. Intrusion

The tooth structure displayed visible areas of higher stress after 4 mm of bone loss on the vestibular and lingual sides (e.g., Figure 4A,B). The intrusion movements displayed the concise areas of maximum shear stress in the external vestibular surface of the radicular dentine/cementum (correlated with the bone loss level), while the moderate resorptive risk yellow and yellow-green areas were shown mainly on the vestibular and lingual external sides (e.g., Figure 4C–J).

The coronal dentine component displayed the highest external shear stress in the intact periodontium, followed by a decrease in the progression of bone loss. Nonetheless, the stress displayed in the internal surface of the dentine/cementum component was mainly moderate (yellow and yellow-green color-coded, mild risk of resorption), with less stress at the pulp chamber and more stress in the vestibular root canal, in all three thirds (e.g., Figure 4E–I). The moderate internal vestibular root canal stress displayed seemed to follow the bone loss level (0 bone loss—cervical third; 4 mm loss—middle and apical third; 8 mm loss—entire apical third). However, at 8 mm of loss, the lingual root canal showed yellow-green traces of shear stress, also prone to mild internal resorption. The quantitative biomechanical display was similar to that shown by the extrusion (Table 2).

**Figure 4 healthcare-11-02622-f004:**
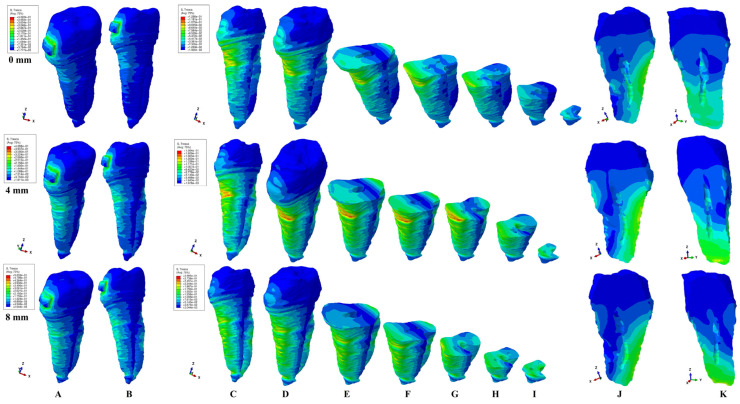
A 0.6 N of intrusion—comparative shear stress display between the tooth structure (**A**,**B**) and the radicular dentine/cementum and coronal dentine components ((**C**,**D**) dentine component, (**E–I**) various horizontal radicular sections in cervical, middle, and apical thirds, (**J**) vertical section of the dentine component viewed from distal side, (**K**) vertical section of the dentine component viewed from lingual side).

### 3.3. Rotation

During the bone loss simulation (especially after 4 mm loss), the tooth structure displayed visible shear stress areas in the middle and apical thirds (e.g., Figure 5A,B). The rotation movements showed on the external surface of the radicular dentine/cementum component the red-orange maximum shear stress (prone to resorption) on medial and distal (lingual after 4 mm loss) sides at the periodontal support level (e.g., Figure 5C,D). The yellow and yellow-green moderate resorption risk areas were more extended (especially after 4 mm of loss) on all sides of the root.

In the coronal dentine, in the intact periodontium, the maximum shear stress was present around and under the bracket with a progressive reduction in stress with bone loss. However, the internal side of the coronal dentine and radicular dentine/cementum components displayed only mild resorptive risk areas (yellow and yellow-green) in both the intact and reduced periodontium, showing a progressive extension along with the periodontal breakdown (e.g., Figure 5E–I). In the intact periodontium, the pulp chamber displayed yellow and yellow-green color-coded stress areas, especially on the vestibular side, correlated with the red-orange shear stress under the bracket position. The color-coded stress progressively decreased in the pulp chamber along with the bone loss (lowest at 8 mm of loss). In the radicular dentine/cementum component, the yellow and yellow-green color-coded stress was displayed in the cervical third (vestibular wall of the pulp chamber and lingual root canal) of the intact periodontium. As the progression of the periodontal breakdown advanced, the moderate stress extended to both root canals in the middle and apical third of the root (highest extension at 8 mm of loss). Quantitatively, the amount of internal shear stress was only half that of the external one, except for the crown, which was comparable. The 8 mm loss of internal stress in the apical and middle third of the radicular dentine/cementum component was 4–5 times higher than in the intact periodontium.

**Figure 5 healthcare-11-02622-f005:**
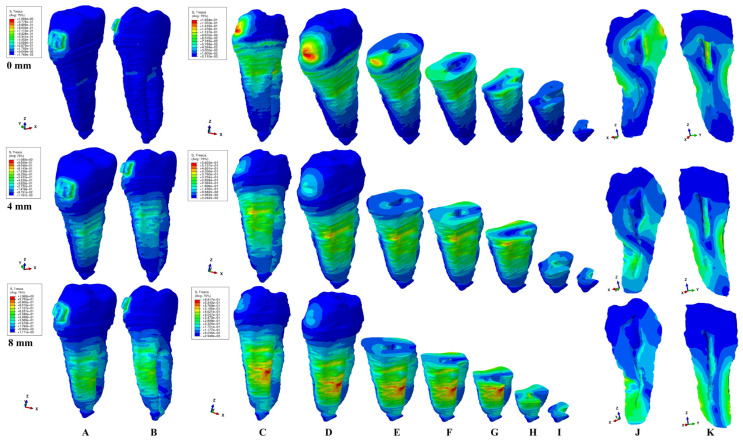
A 0.6 N of rotation—comparative shear stress display between the tooth structure (**A**,**B**) and the radicular dentine/cementum and coronal dentine components ((**C**,**D**) dentine component, (**E**–**I**) various horizontal radicular sections in cervical, middle, and apical thirds, (**J**) vertical section of the dentine component viewed from distal side, (**K**) vertical section of the dentine component viewed from lingual side).

### 3.4. Tipping

In the tooth structure, the red-orange maximum shear stress was more visible from 4 to 8 mm of loss on the lingual side (e.g., Figure 6A,B). After 4 mm of bone loss, the tipping movement (e.g., Figure 6) displayed external red-orange maximum stress areas found around the bone level support on the vestibular and lingual sides. The yellow and yellow-green moderate resorptive risk areas were more surface extended after 4 mm of loss on all external sides of the root. The internal surface of the radicular dentine/cementum component displayed correlated stress areas with the external surface. From 0 to 4 mm of bone loss, the tipping seemed less affected by resorptive risks (only blue-green low-stress areas), while as the periodontal breakdown progressed, the root canals seemed to show yellow and yellow-green mild risks areas in both apical and middle thirds of the root (e.g., Figure 6E–I). Quantitatively, the internal shear stress remained half of the external stress in the apical and middle thirds (except for the dentine crown, which was comparable). The stress increase at 8 mm of bone loss was three times higher for the internal stress in the apical and middle thirds compared with the intact periodontium.

**Figure 6 healthcare-11-02622-f006:**
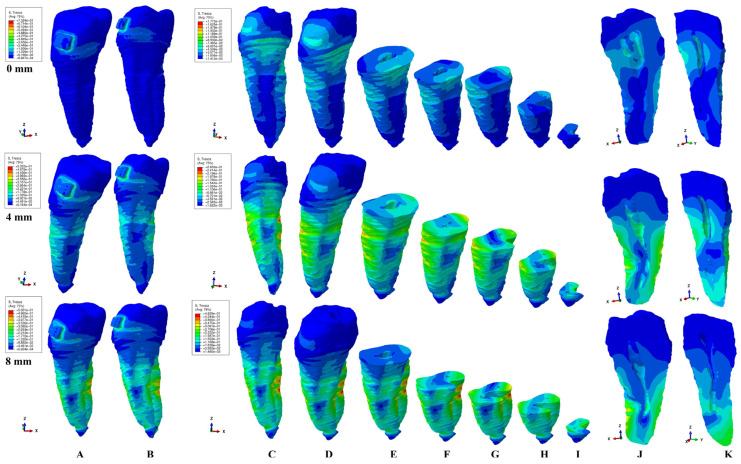
A 0.6 N of tipping—comparative shear stress display between the tooth structure (**A**,**B**) and the radicular dentine/cementum and coronal dentine components ((**C**,**D**) dentine component, (**E**–**I**) various horizontal radicular sections in cervical, middle, and apical thirds, (**J**) vertical section of the dentine component viewed from distal side, (**K**) vertical section of the dentine component viewed from lingual side).

### 3.5. Translation

In the tooth structure, the prone-to-resorption red-orange areas were visible on the lingual, mesial, and distal sides of the root, correlating with the progression of the bone loss (e.g., Figure 7A,B). The red-orange maximum shear stress (e.g., Figure 7C–I) localized areas were shown on the external surface of the radicular dentine/cementum component in both the intact and reduced periodontium, and on the mesial, distal, and lingual sides around the bone support level. The external shear stress distribution correlated with the internal distribution. The internal distribution in the dentine/cementum component from 0 to 4 mm of loss seems to display yellow and yellow-green areas of moderate resorptive risks in the vestibular side/wall of the pulp chamber and lingual root canal, in both the cervical and middle thirds (e.g., Figure 7E–I). After 4 mm of periodontal breakdown, the vestibular root canal seems to display moderate-risk yellow and yellow-green areas in the apical and middle thirds of the root. The coronal dentine component displayed yellow and yellow-green shear stress in the intact periodontium, especially in the vestibular, occlusal, and lingual sides/walls of the pulp chamber, with a progressive decrease of up to 4 mm of loss. From the quantitative point of view, the internal shear stress was half of the external one in both the apical and middle third, and almost comparable for the cervical third and coronal dentine. At 8 mm periodontal breakdown, the stress increase was 3.7–4.9 times in apical and middle thirds when compared with the intact periodontium.

**Figure 7 healthcare-11-02622-f007:**
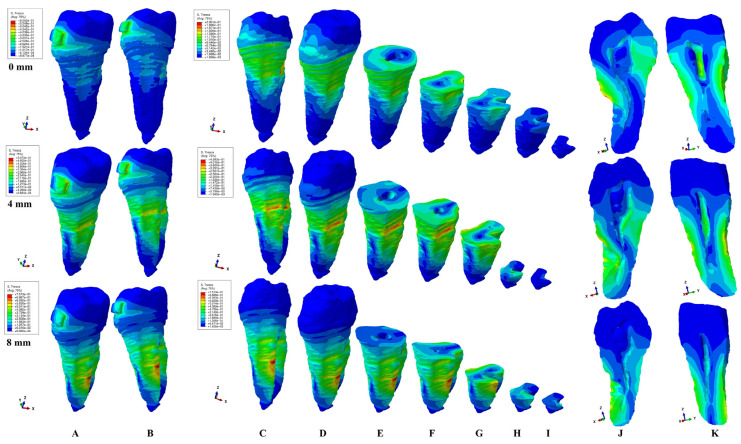
A 0.6 N of translation—comparative shear stress display between the tooth structure (**A**,**B**) and the radicular dentine/cementum and coronal dentine components ((**C**,**D**) dentine component, (**E**–**I**) various horizontal radicular sections in cervical, middle, and apical thirds, (**J**) vertical section of the dentine component viewed from distal side, (**K**) vertical section of the dentine component viewed from lingual side).

The internal and external shear stress complementary display (better visible in the radicular dentine/cementum and coronal dentine components, especially after 4 mm of loss) of localized red-orange maximum stress and yellow and yellow-green moderate stress, suggest that the resorptive risks are influenced by the bone loss levels, due to changes in the biomechanical behavior of the tooth correlated with the periodontal support.

All five orthodontic movements (under both applied forces) showed areas of internal stress located in the root canals, with an extension of the stress areas with the progression of bone loss. In addition, rotation and translation also showed higher stress areas in the pulp chamber in the intact periodontium. Thus, the rotational and translational movements seemed to display moderate resorptive risks in the pulp chamber vestibular wall in the intact periodontium with the risks decreasing with bone loss progression.

A previous study [4] of our team (similar boundary conditions) focused on the maximum amount of shear stress over the pulp and neuro-vascular bundle (Table 3). Thus, by correlating herein tooth shear stress quantitative results (Table 2) with dental pulp–NVB’s shear stress (Table 3), not only a more comprehensive complementary biomechanical view about the absorption–dissipation ability of the tooth appears, but also the relationships with the periodontal breakdown process. For 0.6 N and 1.2 N, the tooth structure absorption–dissipation ability of the shear stress was 98.1–99.97% of all the applied forces up to 8 mm of periodontal breakdown, 0.0114 N/1.14 gf (from 0.6 N) and 0.0228 N/2.28 gf (from 1.2 N), and only 0.46–2.38% reaches the NVB; 0.03–0.24%, the apical middle; 0.04–0.14%, the cervical thirds; and 0.08–0.24%, coronal pulp.

**Figure 8 healthcare-11-02622-f008:**
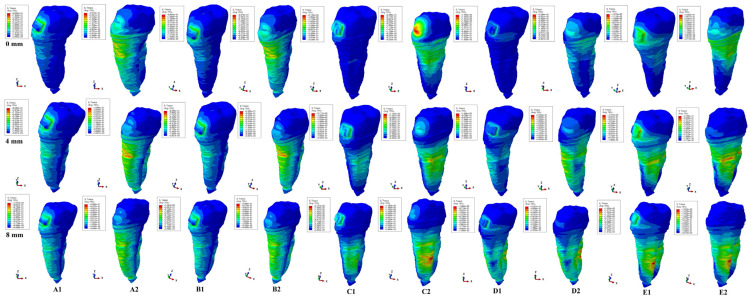
Comparative shear stress display for 1.2 N of intrusion (**A1** tooth, **A2** dentine component), extrusion (**B1** tooth, **B2** dentine component), rotation (**C1** tooth, **C2** dentine component), tipping (**D1** tooth, **D2** dentine component), translation (**E1** tooth, **E2** dentine component).

Biomechanically, the rotation and translation seemed to be the movements prone to produce moderate (yellow and yellow-green color-coded) internal resorptive risk in the root canals (with extension and localization depending on the bone loss level) and in the pulp chamber on the vestibular side/wall (e.g., Figure 3, Figure 4, Figure 5, Figure 6, Figure 7 and Figure 8). Despite the display of stress areas and the quantitative results, under 1.2 N and up to 8 mm bone loss, the intrusion and extrusion seemed less prone to internal resorptive risks. The increase in the quantitative internal stress and the qualitative color-coded stress in the apical third after 4 mm of loss, seemed to suggest that the periodontal breakdown process increased the risks of internal root resorption if the applied force was maintained at the same values as for no bone loss. Nonetheless, 1.2 N of applied force seemed of little resorptive risks in an intact periodontium.

**Table 2 healthcare-11-02622-t002:** Maximum stress average values (KPa) produced by orthodontic forces in the entire tooth structure and in dentine component.

Resorption (mm)			0	1	2	3	4	5	6	7	8
Intrusion	Structure	a	73.51	100.14	126.77	153.40	180.03	200.70	221.36	242.03	262.72
0.6 N/60 gf		m	73.51	100.14	126.77	153.40	180.03	200.70	221.36	242.03	262.72
		c	109.41	135.97	162.52	189.08	215.63	227.40	239.18	250.95	262.72
		C	145.20	162.81	180.42	198.02	215.63	227.40	239.18	250.95	262.72
	Dentin	a	44.18	54.19	64.21	74.22	84.23	88.39	92.56	96.72	100.88
		m	44.18	54.19	64.21	74.22	84.23	88.39	92.56	96.72	100.88
		c	54.72	57.99	61.25	64.52	67.78	69.87	71.97	74.06	76.15
		C	44.18	45.97	47.76	49.54	51.33	57.54	63.74	69.95	76.15
1.2 N/120 gf	Structure	a	147.02	200.28	253.54	306.80	360.05	401.39	442.72	484.05	525.44
		m	147.02	200.28	253.54	306.80	360.05	401.39	442.72	484.05	525.44
		c	218.82	271.93	325.04	378.15	431.25	454.80	478.35	501.90	525.44
		C	290.40	325.62	360.83	396.05	431.25	454.80	478.35	501.90	525.44
	Dentin	a	88.35	108.39	128.41	148.44	168.46	176.79	185.11	193.44	201.75
		m	88.35	108.39	128.41	148.44	168.46	176.79	185.11	193.44	201.75
		c	109.44	115.97	122.50	129.03	135.56	139.75	143.93	148.12	152.30
		C	88.35	91.94	95.51	99.09	102.67	115.07	127.48	139.89	152.30
Extrusion	Structure	a	73.51	100.14	126.77	153.40	180.03	200.70	221.36	242.03	262.72
0.6 N/60 gf		m	73.51	100.14	126.77	153.40	180.03	200.70	221.36	242.03	262.72
		c	109.41	135.97	162.52	189.08	215.63	227.40	239.18	250.95	262.72
		C	145.20	162.81	180.42	198.02	215.63	227.40	239.18	250.95	262.72
	Dentin	a	44.18	54.19	64.21	74.22	84.23	88.39	92.56	96.72	100.88
		m	44.18	54.19	64.21	74.22	84.23	88.39	92.56	96.72	100.88
		c	54.72	57.99	61.25	64.52	67.78	69.87	71.97	74.06	76.15
		C	44.18	45.97	47.76	49.54	51.33	57.54	63.74	69.95	76.15
1.2 N/120 gf	Structure	a	147.02	200.28	253.54	306.80	360.05	401.39	442.72	484.05	525.44
		m	147.02	200.28	253.54	306.80	360.05	401.39	442.72	484.05	525.44
		c	218.82	271.93	325.04	378.15	431.25	454.80	478.35	501.90	525.44
		C	290.40	325.62	360.83	396.05	431.25	454.80	478.35	501.90	525.44
	Dentin	a	88.35	108.39	128.41	148.44	168.46	176.79	185.11	193.44	201.75
		m	88.35	108.39	128.41	148.44	168.46	176.79	185.11	193.44	201.75
		c	109.44	115.97	122.50	129.03	135.56	139.75	143.93	148.12	152.30
		C	88.35	91.94	95.51	99.09	102.67	115.07	127.48	139.89	152.30
Translation	Structure	a	67.94	82.78	97.63	112.47	127.31	236.39	345.46	454.54	563.61
0.6 N/60 gf		m	67.94	156.66	245.38	334.10	465.02	536.60	608.19	679.77	751.35
		c	202.41	257.51	312.62	367.72	422.82	458.02	493.22	528.41	563.61
		C	202.42	236.42	270.41	304.41	338.40	363.43	388.47	413.50	438.53
	Dentin	a	51.43	75.37	99.32	123.26	147.20	157.66	168.12	178.57	189.03
		m	51.43	84.48	117.53	150.57	183.62	200.61	217.60	234.58	251.57
		c	134.07	146.46	158.85	171.23	183.62	200.61	217.60	234.58	251.57
		C	67.94	69.55	71.16	72.77	74.38	87.45	100.51	113.58	126.61
1.2 N/120 gf	Structure	a	135.88	165.57	195.25	224.94	254.61	472.77	690.92	909.07	1127.22
		m	135.88	313.32	490.76	668.20	930.05	1073.21	1216.37	1359.54	1502.69
		c	404.82	515.03	625.23	735.44	845.63	916.04	986.43	1056.83	1127.22
		C	404.83	472.83	540.82	608.81	676.80	726.87	776.93	827.00	877.05
	Dentin	a	102.86	150.75	198.63	246.52	294.41	315.32	336.23	357.15	378.05
		m	102.86	168.96	235.05	301.15	367.24	401.22	435.19	469.17	503.13
		c	268.15	292.92	317.69	342.47	367.24	401.22	435.19	469.17	503.13
		C	135.88	139.10	2871.16	145.54	148.76	174.89	201.02	227.15	253.21
Rotation	Structure	a	90.50	158.53	226.55	294.58	362.60	427.48	492.76	558.04	623.72
0.6 N/60 gf		m	90.50	181.11	271.72	362.32	452.93	540.10	627.27	714.44	801.61
		c	179.20	292.80	406.40	520.00	633.60	653.38	673.17	692.95	712.73
		C	356.62	358.12	359.61	361.11	362.60	383.41	404.22	425.03	445.84
	Dentin	a	57.89	90.82	123.75	156.67	189.60	200.33	211.06	221.79	232.52
		m	57.89	102.48	147.06	191.65	236.23	249.67	263.10	276.54	289.97
		c	127.63	143.12	158.62	174.11	189.60	214.69	239.79	265.88	289.97
		C	127.63	131.50	135.37	139.23	143.10	151.12	159.15	167.17	175.19
1.2 N/120 gf	Structure	a	181.00	317.05	453.10	589.15	725.20	854.96	985.52	1116.08	1247.45
		m	181.00	362.22	543.43	724.65	905.86	1080.20	1254.54	1428.88	1603.23
		c	358.40	585.60	812.80	1040.00	1267.20	1306.77	1346.33	1385.90	1425.46
		C	713.24	716.23	719.22	722.21	725.20	766.82	808.44	850.06	891.67
	Dentin	a	115.78	181.64	247.49	313.35	379.20	400.66	422.12	443.58	465.04
		m	115.78	204.95	294.12	383.29	472.46	499.33	526.20	553.07	579.93
		c	255.26	286.25	317.23	348.22	379.20	429.39	479.57	531.76	579.93
		C	255.26	263.00	270.73	278.47	286.20	302.25	318.29	334.34	350.37
Tipping	Structure	a	61.67	90.71	119.74	148.78	177.81	232.79	287.76	342.74	397.71
0.6 N/60 gf		m	61.67	101.79	141.91	182.03	222.15	299.15	376.14	453.14	530.13
		c	122.61	169.65	216.68	263.72	310.75	332.49	354.23	375.97	397.71
		C	183.65	204.34	225.03	245.72	266.41	299.24	332.06	364.89	397.71
	Dentin	a	45.36	56.23	67.09	77.96	88.82	105.46	122.11	138.75	155.39
		m	45.36	56.23	67.09	77.96	88.82	105.46	122.11	138.75	155.39
		c	60.01	67.21	74.42	81.62	88.82	95.83	102.83	109.84	116.84
		C	45.37	50.78	56.19	61.60	67.01	69.85	72.70	75.54	78.38
1.2 N/120 gf	Structure	a	123.33	181.41	239.48	297.55	355.62	465.57	575.52	685.47	795.43
		m	123.33	203.58	283.82	364.06	444.30	598.29	752.28	906.27	1060.26
		c	245.23	339.29	433.36	527.43	621.50	664.98	708.46	751.94	795.43
		C	367.30	408.68	450.06	491.44	532.81	598.47	664.12	729.77	795.43
	Dentin	a	90.72	112.45	134.18	155.91	177.63	210.93	244.21	277.50	310.77
		m	90.72	112.45	134.18	155.91	177.63	210.93	244.21	277.50	310.77
		c	120.03	134.43	148.83	163.24	177.63	191.65	205.66	219.67	233.68
		C	90.74	101.56	112.38	123.20	134.02	139.71	145.39	151.08	156.77

Structure—stress displayed by the entire tooth structure including bracket. dentin—stress displayed by the dentin component of the tooth structure. a—root apical third, m—root middle third, c—root cervical third, C—crown.

**Table 3 healthcare-11-02622-t003:** Tresca criteria—Maximum stress average values (KPa) displayed in tooth structure, dental pulp and NVB, and % of the tooth quantitative stress that is displayed by the pulp and NVB (absorption–dissipation).

	Resorption (mm)		Apical	Middle	Cervical	Coronal	% Apical	% Middle	% Cervical	% Coronal
Tooth	0	rotation	90.50	90.50	179.20	356.62	100.00	100.00	100.00	100.00
0.6 N/60 gf	8	rotation	623.72	801.61	712.73	445.84	100.00	100.00	100.00	100.00
	0	translation	67.94	67.94	202.41	202.42	100.00	100.00	100.00	100.00
	8	translation	563.61	751.35	563.61	438.53	100.00	100.00	100.00	100.00
	0	tipping	61.67	61.67	122.61	183.65	100.00	100.00	100.00	100.00
	8	tipping	397.71	530.13	397.71	397.71	100.00	100.00	100.00	100.00
	0	intrusion	73.51	73.51	109.41	145.20	100.00	100.00	100.00	100.00
	8	intrusion	262.72	262.72	262.72	262.72	100.00	100.00	100.00	100.00
	0	extrusion	73.51	73.51	109.41	145.20	100.00	100.00	100.00	100.00
	8	extrusion	262.72	262.72	262.72	262.72	100.00	100.00	100.00	100.00
1.2 N/120 gf	0	rotation	181.00	181.00	358.40	713.24	100.00	100.00	100.00	100.00
	8	rotation	1247.45	1603.23	1425.46	891.67	100.00	100.00	100.00	100.00
	0	translation	135.88	135.88	404.82	404.83	100.00	100.00	100.00	100.00
	8	translation	1127.22	1502.69	1127.22	877.05	100.00	100.00	100.00	100.00
	0	tipping	123.33	123.33	245.23	367.30	100.00	100.00	100.00	100.00
	8	tipping	795.43	1060.26	795.43	795.43	100.00	100.00	100.00	100.00
	0	intrusion	147.02	147.02	218.82	290.40	100.00	100.00	100.00	100.00
	8	intrusion	525.44	525.44	525.44	525.44	100.00	100.00	100.00	100.00
	0	extrusion	147.02	147.02	218.82	290.40	100.00	100.00	100.00	100.00
	8	extrusion	525.44	525.44	525.44	525.44	100.00	100.00	100.00	100.00
Pulp–NVB	0	rotation	1.72	0.15	0.15	0.29	1.90	0.17	0.08	0.08
Ref. [3]	8	rotation	6.14	0.57	0.57	1.08	0.98	0.07	0.08	0.24
0.6 N/60 gf	0	translation	1.11	0.11	0.11	0.28	1.63	0.16	0.05	0.14
	8	translation	2.59	0.25	0.25	0.67	0.46	0.03	0.04	0.15
	0	tipping	1.47	0.15	0.15	0.15	2.38	0.24	0.12	0.08
	8	tipping	4.49	0.39	0.39	0.39	1.13	0.07	0.10	0.10
	0	intrusion	1.71	0.15	0.15	0.15	2.33	0.20	0.14	0.10
	8	intrusion	4.34	0.37	0.37	0.37	1.65	0.14	0.14	0.14
	0	extrusion	1.71	0.15	0.15	0.15	2.33	0.20	0.14	0.10
	8	extrusion	4.34	0.37	0.37	0.37	1.65	0.14	0.14	0.14
1.2 N/120 gf	0	rotation	3.45	0.31	0.31	0.58	1.91	0.17	0.09	0.08
	8	rotation	12.29	1.14	1.14	2.15	0.99	0.07	0.08	0.24
	0	translation	2.23	0.22	0.22	0.57	1.64	0.16	0.05	0.14
	8	translation	5.17	0.50	0.50	1.35	0.46	0.03	0.04	0.15
	0	tipping	2.93	0.25	0.25	0.30	2.38	0.20	0.10	0.08
	8	tipping	8.97	0.78	0.78	0.78	1.13	0.07	0.10	0.10
	0	intrusion	3.42	0.30	0.30	0.30	2.33	0.20	0.14	0.10
	8	intrusion	8.68	0.74	0.74	0.74	1.65	0.14	0.14	0.14
	0	extrusion	3.42	0.30	0.30	0.30	2.33	0.20	0.14	0.10
	8	extrusion	8.68	0.74	0.74	0.74	1.65	0.14	0.14	0.14

Apical—apical third, middle—middle third, cervical—cervical third, coronal—coronal third. % apical—% apical third, % middle—% middle third, % cervical—% cervical third, % coronal—% coronal third. tooth—entire tooth structure, NVB—neurovascular bundle.

## 4. Discussion

This finite elements analysis assessed the internal surface resorption risks in the dentine/cementum and coronal dentine components employing the maximum shear stress criteria (to the best of our knowledge being the only one of this type). By simulating the conditions of a gradual horizontal periodontal breakdown, it was possible to study the influence of bone loss over the internal resorptive process during the five orthodontic movements and under 0.6 and 1.2 N.

Since no other studies regarding these issues were found, the correlation of the results herein was performed with the results of external [7,20,28,29,30,31,34] and internal [9] resorption studies, due to the close correlation between the external and internal surface resorptive processes [1,2,3,6,7,14,19,20,34].

The maximum shear stress criteria were able to provide an accurate display of the correlated localized areas of maximum (red-orange) and high (yellow and yellow-green) stress on both the external and internal radicular dentine/cementum and coronal dentine surfaces. These areas are prone to high and moderate risks of the resorptive process, since the resorption seems to be the result of localized stress concentrations [18] due to the force appliance, anatomical morphology, and movement closely correlated with individual susceptibility [1,2,3,6,7,14,19,20,34].

When comparing the results herein with previous studies [7,20,28,29,30,31,34], the first issue that arises is related to the different failure criteria employed in the FEA analysis, which could significantly interfere with the accuracy of the results [8,10,11,21]. The FEA was originally developed for the engineering field, studying materials with micro-architectures less complicated than human tissues and subjected to excessive amounts of force. Each failure criterion was mathematically designed to reproduce and describe the biomechanical behavior of a certain type of material, with limited but visible differences under small, applied forces and movements, but significantly increasing with the amount of force and the amplitude of the movements [8,10,11].

Previous external surface resorption studies [7,20,28,29,30,31,34] largely employed the hydrostatic pressure criterion, specially designed for liquids and gas (i.e., a physical condition where there are no shear stresses), while the analyzed tissues (tooth and periodontium) resemble more ductile materials (with a certain brittle flow mode) that suffer from important shear stresses [3,4,5,8,10,11,12,23,26,27]. The criteria for ductility are the von Mises overall stress (for homogenous materials) and Tresca maximum shear stress (for non-homogenous materials) [3,4,5,8,10,11,12]. FEA comparative studies reported for dental tissues that Tresca and von Mises are adequate, with Tresca providing more accurate results [3,4,5,8,10,11,12]. The main issues regarding the difference between the hydrostatic criteria studies [7,20,28,29,30,31,34] and the Tresca criteria were related to the qualitative stress display, with a more accurate and localized stress area for Tresca [3,4,5,8,10,11,12]. There are also notable quantitative differences, with the hydrostatic studies providing various amounts of stress that sometimes contradict the clinical data [28,29] and/or reporting optimal forces varying from one study to another for the same tooth, movement, and boundary conditions [15,16,17], in contrast to reports by Proffit et al. [13].

The present study (by employing the adequate material-based failure criterion), analyzing the tooth structure, radicular dentine/cementum, and coronal dentine, showed a shear stress display correlated with the bone loss level for both the external and internal prone-to-resorption areas. The external surface radicular dentine/cementum and coronal dentine displayed higher resorptive risks when compared with the internal surface (external surface red-orange and yellow areas vs. internal surface yellow and yellow-green areas), in line with reports regarding the absorption–dissipation ability of the tooth structure [3,4,5,8,10,11,12,23,24,25,26,27]. No significant visible qualitative differences between the 0.6 N and 1.2 N color-coded display areas (about both external and internal surfaces) were seen, so it seems that the only difference consists in a doubling of the amount of quantitative stress display (Table 2).

The resorptive risks seemed to increase with the progression of the bone loss, since most of the maximum shear stress areas of the external surface were displayed around the bone support level and with a visible correspondence of stress display on the internal surface (Figure 3, Figure 4, Figure 5, Figure 6 and Figure 7).

Internal surface moderate resorptive risks (yellow and yellow-green) were visible in both the root canals and radicular and coronal pulp chamber. The biomechanical color-coded display of shear stress seemed to change along with the bone loss for the coronal dentine vs. the radicular dentine/cementum (Figure 3, Figure 4, Figure 5, Figure 6, Figure 7 and Figure 8). If, in the intact periodontium, the color-coded yellow areas are present, especially in the vestibular side/wall of the pulp chamber and cervical third of the radicular dentine/cementum (blue-green in middle and apical third), with the progression of periodontal breakdown, the stress decreases in the pulp chamber (blue-green) and increases (yellow and yellow-green) in the radicular dentine in the middle and apical third, for all five movements and forces (Figure 3, Figure 4, Figure 5, Figure 6, Figure 7 and Figure 8).

Based on the color-coded stress display and quantitative results, the rotational and translational movements seem to be more prone to resorptive risks, while the intrusion and extrusion are less predisposed. It seems that the internal resorptive risks increase with bone loss if the applied force remains unchanged as in the intact periodontium.

It must be emphasized that the two forces 0.6 N and 1.2 N are considered light forces, while, biomechanically, if the amount of force further increases, the results could also change [2,3,4,5,6,8,10,11,13,14], since around 1 N of applied force and under extremely small/reduced displacements/movements, all tissues display linear elasticity (i.e., the higher the force, the more significant the changes appear in the biomechanical behavior). The selection of the amount was chosen due to their frequent use in clinical practice and their employment in previous FEA simulations focusing on PDL and dental pulp and NVB [4,5]. These FEA studies [4,5] reported that 1.2 N could be safely used in an intact periodontium for all movements, while at 8 mm of loss, 0.6 N should not be exceeded.

Only one study assessing the internal resorptive issues was found. Thus, Aslan et al. [9] (300 N occlusal force buccal side oriented, von Mises, single idealized mandibular premolar with root canal filling with gutta/MTA, intact periodontium, 271,837 tetrahedral elements, and 414,930 nodes) reported higher buccal stress than lingual in the cervical and middle thirds of the root, with external surface maximum red-orange overall stress of 74.32–107 MPa (approx. 74,322–107,000 KPa), and high yellow stress of 57.8–83.1 MPa (approx. 57,800–83,100 KPa) and internal surface stress maximum red-orange of 150.5 MPa (approx. 150,500 KPa) cervical, and 381.1 MPa (approx. 381,100 KPa) middle thirds. These results were quantitatively significantly higher than the 29–73.1 MPa reported as maximum shear stress for dentine [24], presuming extremely high risks of resorptive processes, with a large extension on both external and internal surfaces, which would be unusual in daily practice.

The main differences between Aslan et al. [9] and the 3D model study, herein, are related to the fact that the internal resorptions were artificially simulated as cavities of 1.8–3.8 mm in diameter (while the remaining thickness of dentine was 1 mm) with an unusual topographic display, which significantly modifies the biomechanical absorption–dissipation ability of the dental structure correlated with the anatomically inaccurate 3D models (14 times fewer tetrahedral elements).

In our study, in the intact periodontium, the quantitative stress for 0.6 N of intrusion was reported as being 73.5–109.4 KPa on the radicular surface of the tooth structure, and 44.1–54.7 KPa on the internal surface of the radicular dentine/cementum. The qualitative results of the external stress distribution showed a similar display area on the vestibular and lingual sides of the premolar for stress (but a different color-coded display).

It must be emphasized that for more than 1 N of force, the linear elasticity premises are no longer correct and adequate for dental tissue biomechanical behavior. Moreover, von Mises failure criteria (overall stress) were designed for ductile homogenous structures, and tooth and surrounding periodontium do not meet these requirements. Only Tresca (maximum shear stress) criteria were designed for non-homogenous materials, and the linear elasticity assumptions became more and more imprecise as the applied force increased. The difference between the two amounts of force (300 N vs. 0.6 N) also represents two different functional circumstances, maximum masticatory bite load (e.g., 20–120 N average bite load [22]) vs. current practical orthodontic loads, with significantly different biomechanical behavior and tissues reactions (endodontically root canal-treated premolar vs. intact tooth, in our simulation).

Ordinola-Zapata et al. [22], in a review study, reported the quasi-general use of a mix of brittle (maximum principal stress) and ductile (von Mises) failure criteria in the FEA studies of dentine (found in the current research flow), as well as their opinion that dentine is a brittle solid, in contrast to the clinical knowledge, FEA, and in vitro reports about obvious ductility [3,4,5,8,10,11,12,23,24,25,26,27].

From the biomechanical point of view, brittle materials do not show a significant absorption–dissipation ability, rather the energy is not dissipated, and the stress wave propagates determining cracks and fractures (due to limited/no ability of deformation). Moreover, despite the biomechanically acknowledged and reported [8,10,23,26,27] ability of dentine and dentine/cementum to absorb–dissipate stresses (specific to ductile), no studies have approached this issue except ours [8]. Thus, correlation and relationships with an earlier study [4] of our group focusing on the dental pulp–NVB stress (possible because of the employment of similar boundary conditions and physical properties) allowed the quantification of the ability of the dentine to absorb and dissipate the stress reaching the dental pulp (Table 3). In both intact and reduced periodontium, from a total of 0.6 and 1.2 N, the dentine absorbed about 98.1–99.97% of the stress determined by the applied orthodontic force, and only 0.46–2.38% reached the pulpal tissue, which is in line with the expected biomechanical behavior (Table 3). Moreover, this correlation also considered the MHP value that was not exceeded either in PDL [5] nor in pulp–NVB [4] (to reduce the ischemic risks and further periodontal loss) as being as accurate as possible and closer to the reported clinical data.

Field et al. [31], considering the brittleness of dental tissues and employing multiple failure criteria (von Mises, Hydrostatic pressure, and Maximum S1 and Minimum S3 Principal Stress) and 0.35 N/0.5 N of tipping over a 3D mandibular model (23,565–32,812 elements, global element size 1.2 mm, and incisor–canine–premolar), reported that the apical third hydrostatic pressure was two times higher than that of the MHP, and S1 and S3 stresses for PDL were fourteen times higher than that of the MHP, implying a significative number of resorptive processes for the periodontium and root, and extended tissue necrosis for a light orthodontic force, in contrast to clinical knowledge [2,3,4,5,6,8,10,11,13,14]. The main issues in this report (as for most FEA studies) were the material-type assumptions and the anatomical accuracy of the employed models [3,4,5,8,10,11,21].

The FEA resorption studies [7,20,28,29,30,34] related to orthodontic external root resorption in an intact periodontium considered the maximum hydrostatic pressure criterion for their analyses. These reports, which assessed the accuracy of FEA in displaying areas of high-stress resorptive risks and were correlated with the in vivo–in vitro results, found resorptive lacunae, with visible discrepancies between the FEA and clinical data (i.e., the FEA results usually identified the side where the resorption could appear and not the localized points of high stress and pressure, as expected [18]).

Only Zhong et al.’s [7] study showed, for an intact periodontium, a correlated image between the in vivo–in vitro results and FEA external root resorption lacunae (with variable diameter, positions, and extensions varying with the tooth being analyzed), displaying as a pattern the recurrent location of the side of the root (in agreement with herein study). Thus, the idea related to the role of anatomical particularities and individual reactivity in both external and internal resorption seems acceptable and confirmed (but with a need for further studies), as well as the unpredictability of resorption due to the variability of the locations of the points of pressure. Nevertheless, the problem of not accurately identifying the point of pressure but only the side of the root was also clearly visible in this correlation [7].

The Tresca criteria results seem to resemble more those provided by the clinical data [7,20,28,29,34], while the differences between our maximum shear stress and hydrostatic stress [7,28,29] simulations come both from the inadequate selection of the failure criteria and various applied boundary conditions [8,10,11].

Ordinola-Zapata et al. [22], in a review study regarding the employment of the FEA method in the study of treated teeth, emphasized the importance of the correct input data and boundary conditions. The anatomical correctness of the analyzed models is essential, especially in the study of internal and external orthodontic resorption, since the process is highly dependent on the individualities of the root morphology [18] and tissue’s anatomical architecture [1,2,6,7,14,19,20,34]. Thus, a mesh with a high number of elements and nodes and a reduced global element size will provide more anatomical accuracy (herein, 5.06–6.05 million C3D4 tetrahedral elements with a global element size of 0.08–0.116 mm vs. Aslan et al.’s [9] 271,837 tetrahedral elements, 414,930 nodes anatomically idealized premolar, or Field et al.’s [31] mandibular incisor–canine–premolar model of 23,565–32,812 elements, with a global element size 1.2 mm).

The boundary conditions are related to the applied assumptions of anatomical micro-architecture (isotropy, homogeneity, and linear elasticity vs. anisotropy, non-homogeneity, and non-linear elasticity). The ductile failure criteria are specially designed for homogenous (von Mises) and non-homogenous (Tresca) materials, and hydrostatic pressure for liquids (no shear stress).

The linear elasticity assumption is correct only if there are limited movements/displacements and up to 1 N of applied force (as in herein), while as the applied force increases, the non-linear behavior of the anatomical tissues influences the results and becomes increasingly visible (discrepancies between the FEA and clinical data). There are FEA studies [30,32,33] that have analyzed the differences in linearity vs. non-linearity for PDL, reporting on under 1 N of applied force differences of up to 20% (i.e., reduction for non-linearity), and finding that non-linear hydrostatic is the only adequate criteria for PDL. However, the employed failure criteria were that of maximum and minimum principal [32,33] stress (for brittle solids), despite the PDL being of ductile resemblance, von Mises (homogenous ductile), and hydrostatic pressure (liquids/gas) [30], without arguing their choice of criteria; thus, their results should be considered with care.

Isotropic material displays the same properties in all directions, while anisotropy represents a change in properties of the different directions/plans. The difference between these two conditions is extremely small under small loads and movements, but significant if those conditions change. Moreover, almost all FEA studies [9] assumed isotropy, linear elasticity, homogeneity, and perfectly bonded surfaces for all components of the models, except the PDL, where the Ogden hyperelastic model [15,16,17] was applied. The Odgen model was specially designed to describe the non-linear stress–strain behavior of rubber and polymers (non-linearly elastic, isotropic, and incompressible) and some biological tissues (PDL did not accurately match the model due to its specific anatomical micro-architecture).

Thus, Wu et al. [15,16,17], using the hydrostatic stress criteria and assuming homogeneity, isotropy, and linear elasticity (except for PDL–Ogden non-linear hyperelastic) in an intact periodontium, reported an optimal force of 0.28–3.31 N for the intact periodontal ligament of canine, premolar, and lateral incisive, but with significant differences from one study to another for the same tooth (e.g., canine: rotation 1.7–2.1 N [17] and 3.31 N [15]; extrusion 0.38–0.4 N [17] and 2.3–2.6 N [16]; premolar: rotation 2.8–2.9 N [15]), and higher than the clinically accepted light forces of 0.5–1 N/approx. 50–100 gf [2,4,5,6,8,10,11,13,14].

Another critical issue that could interfere with the results of an FEA simulation is related to the used sample size. In the herein simulations, the sample size was nine (nine models originated from nine patients, totaling eighty-one models and 324 simulations), in contrast to the above-mentioned FEA analyses with a sample size of one (one patient and one model). It is accepted that for increasing the accuracy of results, the higher the number of analyzed models, the better the accuracy. Nevertheless, the FEA studies employed only one 3D model (except for our study) due to difficulties in the reconstruction process. The reconstruction process could be performed manually (better anatomical accuracy but extremely difficult and time-consuming, as herein) or automated (less accuracy but also less difficult and time-consuming). The herein models had 28–184 times more elements and a global element size ten times lower than the above-mentioned FEA studies.

Our herein FEA analysis, despite the limited correlation with other internal resorption studies (no orthodontic internal resorption studies were found), showed that the localized areas of potential resorptive risks seemed to vary more with the movement and bone loss level, and less with the amount of applied force. A possible explanation is the fact that both employed forces were light and the movements/displacements were extremely small, while other FEA studies reported a change in the diameter and depth of the resorptive lacunae with the increase in force [7,20,34]. Such a variation could also be possible for our models if a higher force were applied. Nevertheless, previous FEA studies reported that 1.2 N could be safely applied in both an intact periodontium and 0.6 N in 8 mm bone loss; thus, a higher force would increase the risks of further periodontal loss [2,3,4,5,6,8,10,11,13,14]. Moreover, our study prefers the hypothesis that the individuality of each patient correlated with the root morphology, and also plays an important part in the location of the displayed high-pressure areas [18], in agreement with other studies [1,2,6,7,14,19,20,34].

In the absence of other FEA studies, the maximum shear stress criteria seem to offer the highest accuracy for potential resorptive areas, while simulations of external–internal resorption should be developed in a reduced periodontium (often encountered among orthodontic patients). In addition to the above-addressed limits (i.e., failure criteria, boundary conditions, material type, amount of force, and mesh model), it must be emphasized that the FEA analysis, despite being the only possible method for this type of study, does not accurately reproduce clinical conditions; thus, correlations with in vivo–in vitro reports are needed.

## 5. Conclusions

The orthodontic internal surface resorptive risks seem less than that of the external surface.The internal surface resorptive risks seem to increase along with the periodontal loss, especially after 4 mm bone loss.The internal and external surface high points of stress prone to resorptive processes seem to be strictly correlated.The qualitative internal surface stress display seems to be almost similar for the two light forces; the only difference being the doubling of quantitative results.The rotation and tipping movement seems to display a higher internal surface resorptive risk in the coronal dentine pulp chamber in an intact periodontium than the other three movements, decreasing with bone loss.The maximum shear stress criteria seem to supply the accurate localization of high points/areas of pressure prone to resorptive risks.The dentine resemblance to ductile, based on its high absorption–dissipation ability, seems to be correct.Using the Tresca failure criterion, the FEA analysis can supply the predictability of areas to be more prone to resorption much more accurately than other criteria.

## 6. Practical Implications

There is little information (mostly theoretical) about orthodontic internal surface resorption; thus, this study provides a complement to the knowledge, and a better and clearer image, of the biomechanical processes that take place during orthodontic movements under light forces (0.6–1.2 N). The 0–8 mm periodontal breakdown simulation showed a close correlation between the external and internal resorptive process, as well as the localized areas affected by the high points of stress that are more prone to resorption being strictly correlated with the bone loss level. Information that the rotation and tipping movements are prone to higher internal resorptive risks than the other three movements is important for the therapeutic decision, especially if various levels of bone loss are present, as there is a proven correlation between bone loss and an increase in resorptive risks. Important as well is the proven ability of stress absorption–dissipation of the dentine and tooth as a single stand structure. This study supplied what seems to be adequate failure criteria for an analysis of dental tissues under resorption. If the circumstances demand, despite being difficult to employ in everyday practice, in certain complicated situations, the FEA simulation could be employed to anticipate and predict certain areas prone to resorptive risks so that clinical treatment can avoid these risks as much as possible.

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
