# Peer review of "Orthodontic Internal Resorption Assessment in Periodontal Breakdown—A Finite Elements Analysis (Part II)"

_healthcare, 2023, doi:10.3390/healthcare11192622_

Round 1

Reviewer 1 Report

 I particularly appreciate the work done by the authors, what I could suggest are only drafting details. First of all, in line 183, the authors maintain that, due to the restrictive inclusion criteria, the number of patients was reduced to nine. My observation is that  I did not find the exclusion criteria in the text.

         The second observation was related to the limits of the study. I believe that, no matter how well a study was conducted, it has certain limits that we should specify in order to know what we are referring to.          Regarding the style of the references, we cannot find the pages for references 3,4,10,11,12. Different citation styles are also used. Sometimes a period is placed after each first name (A.R), sometimes not, sometimes it is used ; before the title, sometimes not.   I want to underline the fact that my observations did not want to minimize the effort of the authors. I only considered that it would bring a plus from the point of view of understanding the text and the correctness of the writing style. And I apologize once again if the address formula led you to the conclusion that I did not give enough interest to this article.

Author Response

Department of Cariology, Endodontics and Oral Pathology

School of Dental Medicine

University of Medicine and Pharmacy

Ms. Nattarika Peaunbida

Assigned Editor

Healthcare - Second Edition of Innovative Solutions for Oral Healthcare

                                                                                                                            September 8th, 2023

Dear Ms. Nattarika Peaunbida,

Thank you very much for your letter dated September 6th, 2023, with the comments of the reviewers. We have now carefully considered the comments of the reviewers and amended the paper accordingly. All changes are highlighted in red throughout the manuscript and included also below.

Reply to Reviewer #1:

We agree and we thank the reviewer for his/her time and comments. Appropriate changes in the manuscript have by now been made. Please see below and in the manuscript.

Concern of the reviewer:

” Comments and Suggestions for Authors

 I particularly appreciate the work done by the authors, what I could suggest are only drafting details. First of all, in line 183, the authors maintain that, due to the restrictive inclusion criteria, the number of patients was reduced to nine. My observation is that I did not find the exclusion criteria in the text.

The second observation was related to the limits of the study. I believe that, no matter how well a study was conducted, it has certain limits that we should specify in order to know what we are referring to.          

Regarding the style of the references, we cannot find the pages for references 3,4,10,11,12. Different citation styles are also used. Sometimes a period is placed after each first name (A.R), sometimes not, sometimes it is used ; before the title, sometimes not.   I want to underline the fact that my observations did not want to minimize the effort of the authors. I only considered that it would bring a plus from the point of view of understanding the text and the correctness of the writing style. And I apologize once again if the address formula led you to the conclusion that I did not give enough interest to this article.”

Point-by-point response to the reviewer’s comments:

  1. Concern of the reviewer:

“I particularly appreciate the work done by the authors, what I could suggest are only drafting details. First of all, in line 183, the authors maintain that, due to the restrictive inclusion criteria, the number of patients was reduced to nine. My observation is that I did not find the exclusion criteria in the text.”

Our response:

  • We thank the reviewer for his/her concern and comments. We do hope that our changes are according to the reviewer‘s remarks.

Revised text: pg.4 line 197-199

“The exclusion criteria were incomplete arches, malposition teeth, restored teeth, large amount of bone loss and inflamed periodontium.”

  1. Concern of the reviewer:

“The second observation was related to the limits of the study. I believe that, no matter how well a study was conducted, it has certain limits that we should specify in order to know what we are referring to.”

Our response:

  • We thank the reviewer for his/her concern and comments. We do hope that our changes are according to the reviewer‘s remarks.

Revised text: pg.22 lines 733-737

“In addition to the limits above addressed (i.e., failure criteria, boundary conditions, material type, amount of force, mesh model), it must be emphasized that the FEA analysis, despite being the only possible method for this type of study, does not accurately reproduce the clinical conditions, thus correlations with in vivo-in vitro reports is needed.” 

  1. Concern of the reviewer:

“Regarding the style of the references, we cannot find the pages for references 3,4,10,11,12. Different citation styles are also used. Sometimes a period is placed after each first name (A.R), sometimes not, sometimes it is used ; before the title, sometimes not.   I want to underline the fact that my observations did not want to minimize the effort of the authors. I only considered that it would bring a plus from the point of view of understanding the text and the correctness of the writing style. And I apologize once again if the address formula led you to the conclusion that I did not give enough interest to this article.”

Our response:

  • We thank the reviewer for his/her concern and comments. We do hope that our changes are according to the reviewer‘s remarks.

The references were automatically imported via PubMed using the EndnoteX7 software and were inserted in the word document using Intl J Environ Res Pub Health style (provided by EndnoteX7 version, seen to resemble to Healthcare style) since we did not find the Healthcare Style among Endnote options. We do hope to get some help and address this issue during the final editing process.

Reviewer 2 Report

In the present study, the authors utilized Finite Element Analysis (FEA) to assess the accuracy of Maximum Shear Stress criteria (Tresca) in the investigation of orthodontic internal surface resorption and the absorption-dissipation ability of dental tissues.

Topic and title part: The research topic is original

The literature part: The literature is sufficient, and the study is well-conducted.

The discussion part: I suggest improving the Discussion section by adding a paragraph about the limitations of the study. Specifically, I believe that authors should discuss any potential limitations of the analysis performed.

The results part: The results are clear and, overall, the paper is readable. However, the authors should add a section about the limitations of the study, highlighting the primary limitations of the utilized analysis.

Language and Grammar: 
Minor editing of the English language required

Overall, I believe that the present work has been well carried out.

 Minor editing of English language required

Author Response

Department of Cariology, Endodontics and Oral Pathology

School of Dental Medicine

University of Medicine and Pharmacy

Ms. Nattarika Peaunbida

Assigned Editor

Healthcare - Second Edition of Innovative Solutions for Oral Healthcare

                                                                                                                            September 8th, 2023

Dear Ms. Nattarika Peaunbida,

Thank you very much for your letter dated September 6th, 2023, with the comments of the reviewers. We have now carefully considered the comments of the reviewers and amended the paper accordingly. All changes are highlighted in red throughout the manuscript and included also below.

Reply to Reviewer #2:

We agree and we thank the reviewer for his/her time and comments. Appropriate changes in the manuscript have by now been made. Please see below and in the manuscript.

Concern of the reviewer:

” Comments and Suggestions for Authors

In the present study, the authors utilized Finite Element Analysis (FEA) to assess the accuracy of Maximum Shear Stress criteria (Tresca) in the investigation of orthodontic internal surface resorption and the absorption-dissipation ability of dental tissues.

Topic and title part: The research topic is original

The literature part: The literature is sufficient, and the study is well-conducted.

The discussion part: I suggest improving the Discussion section by adding a paragraph about the limitations of the study. Specifically, I believe that authors should discuss any potential limitations of the analysis performed.

The results part: The results are clear and, overall, the paper is readable. However, the authors should add a section about the limitations of the study, highlighting the primary limitations of the utilized analysis.

Language and Grammar:  Minor editing of the English language required

Overall, I believe that the present work has been well carried out.”

Point-by-point response to the reviewer’s comments:

  1. Concern of the reviewer:

“The discussion part: I suggest improving the Discussion section by adding a paragraph about the limitations of the study. Specifically, I believe that authors should discuss any potential limitations of the analysis performed.”

Our response:

  • We thank the reviewer for his/her concern and comments. We do hope that our changes are according to the reviewer‘s remarks.

Revised text: pg.22 lines 733-737

“In addition to the limits above addressed (i.e., failure criteria, boundary conditions, material type, amount of force, mesh model), it must be emphasized that the FEA analysis, despite being the only possible method for this type of study, does not accurately reproduce the clinical conditions, thus correlations with in vivo-in vitro reports is needed.” 

  1. Concern of the reviewer:

“The results part: The results are clear and, overall, the paper is readable. However, the authors should add a section about the limitations of the study, highlighting the primary limitations of the utilized analysis.”

Our response:

  • We thank the reviewer for his/her concern and comments. We do hope that our changes are according to the reviewer‘s remarks.

Revised text: pg.22 lines 733-737

“In addition to the limits above addressed (i.e., failure criteria, boundary conditions, material type, amount of force, mesh model), it must be emphasized that the FEA analysis, despite being the only possible method for this type of study, does not accurately reproduce the clinical conditions, thus correlations with in vivo-in vitro reports is needed.” 

  1. Concern of the reviewer:

“Language and Grammar:  Minor editing of the English language required.”

Our response:

  • We thank the reviewer for his/her concern and comments. We do hope that our changes are according to the reviewer‘s remarks.

Revised text: entire manuscript

Reviewer 3 Report

1. Please Rewrite your entire manuscript according to PRILE guidelines. 
2. please provide the details on sample size calculation, power and effect size details. 

3. please provide the sample size details and p values in results section . 
4. inclusion and exclusion criteria are unclear please reframe methodology according to PRILE guidelines. 
5. introduction and discussion can be concise. Overall the manuscript lacks flow.it’s hard to follow. 
6. Conclusion is not crisp and clear. 

Author Response

Department of Cariology, Endodontics and Oral Pathology

School of Dental Medicine

University of Medicine and Pharmacy

Ms. Nattarika Peaunbida

Assigned Editor

Healthcare - Second Edition of Innovative Solutions for Oral Healthcare

                                                                                                                            September 8th, 2023

Dear Ms. Nattarika Peaunbida,

Thank you very much for your letter dated September 6th, 2023, with the comments of the reviewers. We have now carefully considered the comments of the reviewers and amended the paper accordingly. All changes are highlighted in red throughout the manuscript and included also below.

Reply to Reviewer #3:

We agree and we thank the reviewer for his/her time and comments. Appropriate changes in the manuscript have by now been made. Please see below and in the manuscript.

Concern of the reviewer:

” Comments and Suggestions for Authors

  1. Please Rewrite your entire manuscript according to PRILE guidelines. 
    2. please provide the details on sample size calculation, power and effect size details. 
  2. please provide the sample size details and p values in results section . 
    4. inclusion and exclusion criteria are unclear please reframe methodology according to PRILE guidelines. 
    5. introduction and discussion can be concise. Overall the manuscript lacks flow.it’s hard to follow. 
    6. Conclusion is not crisp and clear.”

Point-by-point response to the reviewer’s comments:

  1. Concern of the reviewer:

“Please Rewrite your entire manuscript according to PRILE guidelines.”

Our response:

  • We thank the reviewer for his/her concern and comments.

With all due respect The PRILE protocol refers to Preferred Reporting Items for Laboratory studies in Endodontology (PRILE) (endodontic studies regarding several teeth with various form of treatments that are studied for comparing the treatment accuracy). We must emphasize that our study is a Finite Element Study and not an endodontic study. There are no significant differences regarding the rules of PRILE and other types of scientific studies, thus we do not consider that our study should be re-written.

Manuscript pg.22 lines:756-772

“There is little information (mostly theoretical) about the orthodontic internal surface resorption, thus this study provides a complement to the knowledge and better and clear image of the biomechanical processes that take place during the orthodontic movements under the light forces (0.6-1.2 N). The 0-8 mm periodontal breakdown simulation shown the close correlation between the external and internal resorptive process, as well as the localized areas affected by the high points of stresses more prone to resorption strictly correlated with the bone loss level. Information that the rotation and tipping movements are prone to higher internal resorptive risks than the other three movements are important for the therapeutic decision especially if various levels of bone loss are present since the proven correlation between the bone loss with the increase of resorptive risks. Important as well is the proven ability of stress absorption-dissipation of the dentine and tooth as a single stand structure. This study supplied what seems to be the adequate failure criteria for the analysis of dental tissues under resorption. If the circumstances demand, despite being difficult to be employed in everyday practice, for certain complicated situations the FEA simulation could be employed to anticipate and predict certain areas prone to resorptive risks so that clinical treatment can avoid this risk as much as possible.”

  1. Concern of the reviewer:

“please provide the details on sample size calculation, power and effect size details.

 please provide the sample size details and p values in results section”

Our response:

  • We thank the reviewer for his/her concern and comments. We do hope that our changes are according to the reviewer‘s remarks.

Revised text: pg.4 lines 185-205

“The current FEA analysis represents a stepwise study of a more comprehensive re-search (clinical protocol nr.158/02.04.2018) of the biomechanical behavior of the tooth and surrounding support tissues under orthodontic movements during the horizontal periodontal breakdown [3-5, 8, 10, 11].

This analysis involved a number of 324 simulations on eighty-one anatomically correct 3D models from nine patients (mean age of 29.81 ± 1.45 years, 4 males and 5 females, oral informed consent).

This research examined a larger number of patients, but a rather restrictive inclusion criteria reduced their number to nine. These inclusion criteria were chosen for reducing the biomechanical uncertainties (i.e., intact mandibular arch, no malposition, intact teeth, no root canals endodontic treatment, no filling, non-inflamed periodontium, moderate/reduce bone loss in the cervical third, orthodontic treatment, regular periodic checks). The exclusion criteria were incomplete arches, malposition teeth, restored teeth, large amount of bone loss and inflamed periodontium. Thus, the sample size was nine (nine patients; nine models/patient; a total of 81 3D models and 324 FEA simulations), opposingly to current FEA studies analyzing a sample size of one (one model of one patient and few FEA simulations) because of the 3D model creation. It must be emphasized that most of previous FEA studies [7, 9, 15-17, 20, 22, 23, 26-34] used for comparation with herein results used a sample size of one (one patient and one model) excepting our earlier [3-5, 8, 10, 11] (a sample size of nine).” 

With all due respect we must emphasize that the FEA studies do not use a large sample size since (usually only one model of one patient) the method allows a wide range of simulation by changing the boundary conditions. However, in order to enhance the accuracy of our results we used nine patients, with nine models, that were modify according to our specific needs obtaining a total of 81 models and a total of 324 simulations. We must add that the reconstruction of a FEA model, especially if is manually reconstructed demands a lot of time and computing power, this being the reason that in current literature flow the sample size for FEA studies is one.

  1. Concern of the reviewer:

“inclusion and exclusion criteria are unclear please reframe methodology according to PRILE guidelines.”

Our response:

  • We thank the reviewer for his/her concern and comments. We do hope that our changes are according to the reviewer‘s remarks.

Revised text: pg.4 lines 185-205

“The current FEA analysis represents a stepwise study of a more comprehensive re-search (clinical protocol nr.158/02.04.2018) of the biomechanical behavior of the tooth and surrounding support tissues under orthodontic movements during the horizontal periodontal breakdown [3-5, 8, 10, 11].

This analysis involved a number of 324 simulations on eighty-one anatomically correct 3D models from nine patients (mean age of 29.81 ± 1.45 years, 4 males and 5 females, oral informed consent).

This research examined a larger number of patients, but a rather restrictive inclusion criteria reduced their number to nine. These inclusion criteria were chosen for reducing the biomechanical uncertainties (i.e., intact mandibular arch, no malposition, intact teeth, no root canals endodontic treatment, no filling, non-inflamed periodontium, moderate/reduce bone loss in the cervical third, orthodontic treatment, regular periodic checks). The exclusion criteria were incomplete arches, malposition teeth, restored teeth, large amount of bone loss and inflamed periodontium. Thus, the sample size was nine (nine patients; nine models/patient; a total of 81 3D models and 324 FEA simulations), oppos-ingly to current FEA studies analyzing a sample size of one (one model of one patient and few FEA simulations) because of the 3D model creation. It must be emphasized that most of previous FEA studies [7, 9, 15-17, 20, 22, 23, 26-34] used for comparation with herein results used a sample size of one (one patient and one model) excepting our earlier [3-5, 8, 10, 11] (a sample size of nine).” 

With all due respect we must emphasize that the FEA studies are different than endodontic studies (i.e., the reference to PRILE protocol). For our study the need was of intact arches and teeth (in order to obtain accurate anatomical mesh models), and with limited bone loss. The inclusion and exclusion criteria emphasized that need.

  1. Concern of the reviewer:

“introduction and discussion can be concise. Overall the manuscript lacks flow.it’s hard to follow.”

Our response:

  • We thank the reviewer for his/her concern and comments. We do hope that our changes are according to the reviewer‘s remarks.

Manuscript pg.22 lines:756-772

“There is little information (mostly theoretical) about the orthodontic internal surface resorption, thus this study provides a complement to the knowledge and better and clear image of the biomechanical processes that take place during the orthodontic movements under the light forces (0.6-1.2 N). The 0-8 mm periodontal breakdown simulation shown the close correlation between the external and internal resorptive process, as well as the localized areas affected by the high points of stresses more prone to resorption strictly correlated with the bone loss level. Information that the rotation and tipping movements are prone to higher internal resorptive risks than the other three movements are important for the therapeutic decision especially if various levels of bone loss are present since the proven correlation between the bone loss with the increase of resorptive risks. Important as well is the proven ability of stress absorption-dissipation of the dentine and tooth as a single stand structure. This study supplied what seems to be the adequate failure criteria for the analysis of dental tissues under resorption. If the circumstances demand, despite being difficult to be employed in everyday practice, for certain complicated situations the FEA simulation could be employed to anticipate and predict certain areas prone to resorptive risks so that clinical treatment can avoid this risk as much as possible.”

This manuscript is the first research of this type assessing the orthodontic internal resorption during periodontal breakdown. Moreover, no other study employed the Tresca failure criteria for the finite element analysis of internal resorption in the search of a single most accurate criteria for the study of bone and dental tissues (tooth and surrounding periodontium included). There are multiple concepts (with correlations and relationships) approached in the introduction and discussion section that needed to be presented for familiarizing the reader (both clinician and researcher) for a better understanding of the study. Since the multitude of data that needed to be addressed and correlated with our results, we tried very hard to provide a scientifical, logical and coherent flow as best as possible. Thus, with all due respect we strongly disagree with reviewer remarks. However, for improving the flow we change the manuscript. We do hope that these modifications will improve the manuscript.  

  1. Concern of the reviewer:

“Conclusion is not crisp and clear.”

Our response:

  • We thank the reviewer for his/her concern and comments. We do hope that our changes are according to the reviewer‘s remarks.
  1. Revised text:22 lines 738-755

“5. Conclusions

  1. The orthodontic internal surface resorptive risks seemed less than that of the external surface.
  2. The internal surface resorptive risks seemed to increase along with the periodontal loss especially after 4 mm bone loss.
  3. The internal and external surface high points of stress prone to resorptive processes seemed to be strictly correlated.
  4. The qualitative internal surface stress display seemed to be almost similar for the two light forces, the only difference being the doubling of quantitative results.
  5. The rotation and tipping movement seemed to display a higher internal surface resorptive risk in the coronal dentine pulp chamber in intact periodontium than the other three movements, decreasing with bone loss.
  6. Maximum shear stress criteria seem to supply accurate localization of high points/areas of pressure prone to resorptive risks.
  7. The dentine resemblance to ductile based on its high absorption-dissipation ability seems to be correct.
  8. Using Tresca failure criterion, the FEA analysis can supply predictability of areas more prone to resorption, much more accurately than other criteria.”

With all due respect we must emphasize that the FEA studies are different than endodontic studies (i.e., the reference to PRILE protocol), thus the conclusions for endodontic studies could be sharp (e.g., YES/NO type). However, the FEA studies by analyzing color coded projections and a large quantity of numerical data present the results and conclusions under another form (present in all FEA studies from the literature, herein included). The conclusions of this study addressed all issues presented in the aims and discussions sections, under a coherent and logical form in order to inform and provide a better understanding of the subject.

Reviewer 4 Report

According to this manuscript, I would like to express my thanks to the authors for their efforts; it needs a minor revision before evaluating the possibility of publication. I would like to pay attention to the following comments:

  • In the introduction, authors should write briefly about Tresca theory of failure.
  • The problem question should be properly addressed in the introduction section.
  • The introduction section is too long need to be concise.
  • The null hypothesis should be added at the end of the introduction section and in the discussion.
  • In the methodology, why you decide the sample size to be nine?
  • In the methodology, why you cause of applying load of 0.6 N and 1.2 N?
  • Line#715 (the maximum shar stress criteria) need correction.
  • Some references need to be updated (within the last 5 years).

Author Response

Department of Cariology, Endodontics and Oral Pathology

School of Dental Medicine

University of Medicine and Pharmacy

Ms. Nattarika Peaunbida

Assigned Editor

Healthcare - Second Edition of Innovative Solutions for Oral Healthcare

                                                                                                                            September 8th, 2023

Dear Ms. Nattarika Peaunbida,

Thank you very much for your letter dated September 6th, 2023, with the comments of the reviewers. We have now carefully considered the comments of the reviewers and amended the paper accordingly. All changes are highlighted in red throughout the manuscript and included also below.

Reply to Reviewer #4:

We agree and we thank the reviewer for his/her time and comments. Appropriate changes in the manuscript have by now been made. Please see below and in the manuscript.

Concern of the reviewer:

” Comments and Suggestions for Authors

According to this manuscript, I would like to express my thanks to the authors for their efforts; it needs a minor revision before evaluating the possibility of publication. I would like to pay attention to the following comments:

  • In the introduction, authors should write briefly about Tresca theory of failure.
  • The problem question should be properly addressed in the introduction section.
  • The introduction section is too long need to be concise.
  • The null hypothesis should be added at the end of the introduction section and in the discussion.
  • In the methodology, why you decide the sample size to be nine?
  • In the methodology, why you cause of applying load of 0.6 N and 1.2 N?
  • Line#715 (the maximum shar stress criteria) need correction.
  • Some references need to be updated (within the last 5 years).”

Point-by-point response to the reviewer’s comments:

  1. Concern of the reviewer:

“In the introduction, authors should write briefly about Tresca theory of failure.”

Our response:

  • We thank the reviewer for his/her concern and comments. We do hope that our changes are according to the reviewer‘s remarks.

Revised text: pg.3 line 148-150

“Tresca failure criteria states that failure occurs in a material when the absolute value of maximum shear stress reaches the shear stress at yielding and is consistent with failure being independent of hydrostatic stress.”

  1. Concern of the reviewer:

“The problem question should be properly addressed in the introduction section.”

Our response:

  • We thank the reviewer for his/her concern and comments. We do hope that our changes are according to the reviewer‘s remarks.
  1. Revised text: pg.3 lines 137-138

“Moreover, no information about the amount of safely applied orthodontic force to avoid orthodontic resorption or absorption-dissipation ability were found.” 

  1. Revised text:4 lines 180-184

 “This FEA analysis aimed to investigate the areas more exposed to risks of orthodontic internal resorption during 0-8 mm of periodontal breakdown under 0.6 N/60 gf and 1.2 N/120 gf and under various movements, as well as the absorption-dissipation ability of the tooth. Additionally, if there are significant differences between the biomechanical behavior of the two forces was also assessed.”

 Due to vastity and complexity of the investigated issue we tried to present as concise as possible all available data strictly correlated with aims of the study.

  1. Concern of the reviewer:

“The introduction section is too long need to be concise.”

Our response:

  • We thank the reviewer for his/her concern and comments. We do hope that our changes are according to the reviewer‘s remarks.

We agree with the reviewer that the introduction is long. However, this study being the first of its kind, we considered that is in the benefit of reader (for an easier reading and understanding) to be familiarized with all available data regarding the addressed issues. When presenting the available information regarding this subject we tried to be as concise as possible and to reduce it to a minimum.

  1. Concern of the reviewer:

“-The null hypothesis should be added at the end of the introduction section and in the discussion.”

Our response:

  • We thank the reviewer for his/her concern and comments. We do hope that our changes are according to the reviewer‘s remarks.

Revised text: pg.4 lines 180-184

 “This FEA analysis aimed to investigate the areas more exposed to risks of orthodontic internal resorption during 0-8 mm of periodontal breakdown under 0.6 N/60 gf and 1.2 N/120 gf and under various movements, as well as the absorption-dissipation ability of the tooth. Additionally, if there are significant differences between the biomechanical behavior of the two forces was also assessed.”

Revised text: pg.18 lines 541-555

“The present study (by employing the adequate material-based failure criterion) analyzing the tooth structure, radicular dentine-cementum and coronal dentine showed a shear stress display correlated with the bone loss level for both external and internal prone to resorption areas. The external surface radicular dentine-cementum and coronal dentine displayed a higher resorptive risks when compared with the internal surface (external surface red-orange and yellow areas vs. internal surface yellow and yellow-green areas), in line with the reports regarding the absorption-dissipation ability of the tooth structure [3-5, 8, 10-12, 23-27]. No significative visible qualitative differences between the 0.6 N and 1.2 N color coded display areas (about both external and internal surfaces) were seen, seeming that the only difference consists in a doubling of the amount of quantitative stress display (Table 2).

The resorptive risks seemed to increase with the progression of the bone loss, since the most of the maximum shear stress areas of the external surface were displayed around the bone support level and with a visible correspondence of stress display on the internal surface (Figures 3-7).”

  1. Concern of the reviewer:

“In the methodology, why you decide the sample size to be nine?”

Our response:

  • We thank the reviewer for his/her concern and comments. We do hope that our changes are according to the reviewer‘s remarks.

Revised text: pg.4 lines 193-205

“This research examined a larger number of patients, but a rather restrictive inclusion criteria reduced their number to nine. These inclusion criteria were chosen for reducing the biomechanical uncertainties (i.e., intact mandibular arch, no malposition, intact teeth, no root canals endodontic treatment, no filling, non-inflamed periodontium, moderate/reduce bone loss in the cervical third, orthodontic treatment, regular periodic checks). The exclusion criteria were incomplete arches, malposition teeth, restored teeth, large amount of bone loss and inflamed periodontium. Thus, the sample size was nine (nine patients; nine models/patient; a total of 81 3D models and 324 FEA simulations), opposingly to current FEA studies analyzing a sample size of one (one model of one patient and few FEA simulations) because of the 3D model creation. It must be emphasized that most of previous FEA studies [7, 9, 15-17, 20, 22, 23, 26-34] used for comparation with herein results used a sample size of one (one patient and one model) excepting our earlier [3-5, 8, 10, 11] (a sample size of nine).”

  1. Concern of the reviewer:

“In the methodology, why you cause of applying load of 0.6 N and 1.2 N?.”

Our response:

  • We thank the reviewer for his/her concern and comments. We do hope that our changes are according to the reviewer‘s remarks.

Revised text: pg.6 lines 250-255

“The applied forces at the bracket level (e.g., Figure 1) were of 0.6 N/approx. 60 gf and 1.2 N/approx. 120 gf, simulating five orthodontic movements (extrusion, intrusion, tipping, rotation, and translation). Those forces were chosen not only because are often used in clinical practice but also to establish correlations with two previous analyses [4, 5, 8] of PDL and dental pulp and NVB improving thus the knowledge regarding the biomechanical behavior of tooth subjected to periodontal breakdown.”

  1. Concern of the reviewer:
  • “Line#715 (the maximum shar stress criteria) need correction.”

Our response:

  • We thank the reviewer for his/her concern and comments. We do hope that our changes are according to the reviewer‘s remarks.

Revised text: pg.22 line 730

“In the absence of other FEA studies, the maximum shear stress criteria seem to offer ….”

  1. Concern of the reviewer:

“Some references need to be updated (within the last 5 years).”

Our response:

  • We thank the reviewer for his/her concern and comments. We do hope that our changes are according to the reviewer‘s remarks.

This manuscript is the first research of this type assessing the biomechanical behavior of bone structure during periodontal breakdown. Moreover, no other study employed multiple failure criteria for the finite element analysis of bone in the search of a single most accurate criteria for the study of bone and dental tissues (tooth and surrounding periodontium included). There are multiple concepts (with correlations and relationships) approached in the introduction section that needed to be presented for familiarizing the reader (both clinician and researcher) for a better understanding of the study. Regarding the old references, it must be emphasized that there is only a little data available in the research flow regarding herein subject, so for a better support of the explanations we needed to use whatever resources were available.

Reviewer 5 Report

I have read your manuscript entitled "Orthodontic Internal Resorption Assessment in Periodontal Breakdown - A Finite Elements Analysis (Part II)".

I have appreciated your manuscript and I enjoyed reading your paper. It was well-organized and easy to follow. Your arguments were clear and persuasive. The research is thorough, the writing is clear and concise, and the conclusions are well-supported. The writing is engaging and the ideas are stimulating.

However, in order to improve your Introduction, it would be important to add some sources as following:

Diagnosis and Management of Mandibular crowding:
   https://pubmed.ncbi.nlm.nih.gov/37240944/

Normofunctional forces analised by FEA:
   https://pubmed.ncbi.nlm.nih.gov/36518139/

Author Response

Department of Cariology, Endodontics and Oral Pathology

School of Dental Medicine

University of Medicine and Pharmacy

Ms. Nattarika Peaunbida

Assigned Editor

Healthcare - Second Edition of Innovative Solutions for Oral Healthcare

                                                                                                                            September 8th, 2023

Dear Ms. Nattarika Peaunbida,

Thank you very much for your letter dated September 6th, 2023, with the comments of the reviewers. We have now carefully considered the comments of the reviewers and amended the paper accordingly. All changes are highlighted in red throughout the manuscript and included also below.

Reply to Reviewer #5:

We agree and we thank the reviewer for his/her time and comments. Appropriate changes in the manuscript have by now been made. Please see below and in the manuscript.

Concern of the reviewer:

” Comments and Suggestions for Authors

I have read your manuscript entitled "Orthodontic Internal Resorption Assessment in Periodontal Breakdown - A Finite Elements Analysis (Part II)".

I have appreciated your manuscript and I enjoyed reading your paper. It was well-organized and easy to follow. Your arguments were clear and persuasive. The research is thorough, the writing is clear and concise, and the conclusions are well-supported. The writing is engaging and the ideas are stimulating.

However, in order to improve your Introduction, it would be important to add some sources as following:

Diagnosis and Management of Mandibular crowding:
   https://pubmed.ncbi.nlm.nih.gov/37240944/

Normofunctional forces analised by FEA:
   https://pubmed.ncbi.nlm.nih.gov/36518139/.”

Point-by-point response to the reviewer’s comments:

  1. Concern of the reviewer:

“However, in order to improve your Introduction, it would be important to add some sources as following:

Diagnosis and Management of Mandibular crowding:
   https://pubmed.ncbi.nlm.nih.gov/37240944/.”

Our response:

  • We thank the reviewer for his/her concern and comments. We do hope that our changes are according to the reviewer‘s remarks.

We read the suggested reference (Mandibular Crowding: Diagnosis and Management-A Scoping Review), however its aim was “To identify relevant studies investigating the most common possible treatments for mandibular dental crowding”.

Since our subject is different:” This FEA analysis aimed to investigate the areas more exposed to risks of orthodontic internal resorption during 0-8 mm of periodontal breakdown under 0.6 N/60 gf and 1.2 N/120 gf and under various movements, as well as the absorption-dissipation ability of the tooth. Additionally, if there are significant differences between the biomechanical behavior of the two forces was also assessed”, we consider that this reference will not significantly improve our manuscript.

  1. Concern of the reviewer:

“However, in order to improve your Introduction, it would be important to add some sources as following:

Normofunctional forces analised by FEA:
   https://pubmed.ncbi.nlm.nih.gov/36518139/.”

Our response:

  • We thank the reviewer for his/her concern and comments. We do hope that our changes are according to the reviewer‘s remarks.

We read the suggested reference (FEA analysis of Normo-functional forces on periodontal elements in different angulations), however its aim was “Therefore, it is of interest to assess the effect of normal occlusal force on periodontal ligament in different angulations”, and under forces of 100-150 N using maximum principal stress and von Mises stress.

Since our subject is different:” This FEA analysis aimed to investigate the areas more exposed to risks of orthodontic internal resorption during 0-8 mm of periodontal breakdown under 0.6 N/60 gf and 1.2 N/120 gf and under various movements, as well as the absorption-dissipation ability of the tooth. Additionally, if there are significant differences between the biomechanical behavior of the two forces was also assessed”, we consider that this reference will not significantly improve our manuscript (since we used Tresca criteria which is more accurate than the criteria used in above research, while the applied forces were light). However, this suggested research is of interest for another manuscript that we prepare, where it will be used as reference.

Round 2

Reviewer 3 Report

Now the study can be accepted 

Author Response

Department of Cariology, Endodontics and Oral Pathology

School of Dental Medicine

University of Medicine and Pharmacy

Ms. Nattarika Peaunbida

Assistant Editor

Healthcare      

Special Issue - Second Edition of Innovative Solutions for Oral Healthcare      

                                                                                                                                 Sept 16th, 2023

Dear Ms. Nattarika Peaunbida,

Thank you very much for your letter dated September 15th, 2023, with the comments of the reviewers. We have now carefully considered the comments of the reviewers and amended the paper accordingly. All changes are highlighted in red throughout the manuscript and included also below.

Reply to Reviewer #3:

We agree and we thank the reviewer for his/her time and comments. Appropriate changes in the manuscript have by now been made. Please see below and in the manuscript.

Concern of the reviewer:

” Comments and Suggestions for Authors

Now the study can be accepted.”

Our response:

  • We thank the reviewer for his/her concern and comments.

Reviewer 5 Report

Dear Authors,

I have detected several auto-citations and I strongly  believe in the modifications I suggested

Author Response

Department of Cariology, Endodontics and Oral Pathology

School of Dental Medicine

University of Medicine and Pharmacy

Ms. Nattarika Peaunbida

Assistant Editor

Healthcare      

Special Issue - Second Edition of Innovative Solutions for Oral Healthcare      

                                                                                                                                 Sept 17th, 2023

Dear Ms. Nattarika Peaunbida,

Thank you very much for your letter dated September 15th, 2023, with the comments of the reviewers. We have now carefully considered the comments of the reviewers and amended the paper accordingly. All changes are highlighted in red throughout the manuscript and included also below.

Reply to Reviewer #5:

.Appropriate changes in the manuscript have by now been made. Please see below and in the manuscript.

Concern of the reviewer:

” Comments and Suggestions for Authors - second stage of review

Dear Authors,

I have detected several auto-citations and I strongly  believe in the modifications I suggested.”

“” Comments and Suggestions for Authors
- first stage of review

I have read your manuscript entitled "Orthodontic Internal Resorption Assessment in Periodontal Breakdown - A Finite Elements Analysis (Part II)".

I have appreciated your manuscript and I enjoyed reading your paper. It was well-organized and easy to follow. Your arguments were clear and persuasive. The research is thorough, the writing is clear and concise, and the conclusions are well-supported. The writing is engaging and the ideas are stimulating.

However, in order to improve your Introduction, it would be important to add some sources as following:

Diagnosis and Management of Mandibular crowding:
   https://pubmed.ncbi.nlm.nih.gov/37240944/

Normofunctional forces analised by FEA:
   https://pubmed.ncbi.nlm.nih.gov/36518139/.”

Our response:

  • We thank the reviewer for his/her concern and comments.

Herein manuscript is a part pf a larger step-by-step study, entirely new, aiming to find the best FEA failure criteria to investigate the biomechanical behaviour of tooth and surrounding periodontium components under orthodontic forces and during periodontal breakdown. All cited references (auto-citations) are important for the benefit of the reader (a better understanding of the subject) and presents steps of the research that are essentials to be read since each of them show a new approach of the problem. All these studies are relevant for the study since they are related to this manuscript.

                     Pg. 4 lines:186-189

“The current FEA analysis represents a stepwise study of a more comprehensive re-search (clinical protocol nr.158/02.04.2018) of the biomechanical behavior of the tooth and surrounding support tissues under orthodontic movements during the horizontal perio-dontal breakdown [3-5, 8, 10, 11].”

Regarding the previous suggested modifications during the first stage of the review process, the suggestions had been implemented.

Modified text: pg. 3 lines 141-142

“…without any correlation with the type of analyzed material [35], MHP, ischemic risks, and reporting…”

                         Pg. 4 lines:157-158

“…with lesser importance to the amount of the applied force [35, 36]…”

                          Pg. 18, line 516

“…morphology and movement closely corelated with the individual susceptibility [1-3, 6, 7, 14, 19, 20, 34-36].”

                    Pg. 26 lines: 886-891

“ 35. Shetty, B.; Fazal, I.; Khan, S. F., FEA analysis of Normofunctional forces on periodontal elements in different angulations. Bioinformation 2022, 18, (3), 245-250.

  1. Patano, A.; Malcangi, G.; Inchingolo, A. D.; Garofoli, G.; De Leonardis, N.; Azzollini, D.; Latini, G.; Mancini, A.; Carpentiere, V.; Laudadio, C.; Inchingolo, F.; D'Agostino, S.; Di Venere, D.; Tartaglia, G. M.; Dolci, M.; Dipalma, G.; Inchingolo, A. M., Mandibular Crowding: Diagnosis and Management-A Scoping Review. Journal of personalized medicine 2023, 13, (5).”
